# Complex de novo structural variants are an underestimated cause of rare disorders

Hyunchul Jung [1] ✉, Tsun-Po Yang[1], Susan Walker [2], Petr Danecek[1], O. Isaac Garcia-Salinas [1], Matthew D. C. Neville [1], Joseph Christopher[1,3,4], Isidro Cortés-Ciriano [1,5], Helen Firth[1,3], Aylwyn Scally [6], Matthew Hurles [1], Peter Campbell [1] & Raheleh Rahbari [1] ✉

Complex de novo structural variants (dnSVs) are crucial genetic factors in rare disorders, yet their prevalence and characteristics in rare disorders remain poorly understood. Here, we conduct a comprehensive analysis of whole-genome sequencing data of 12,568 families, including 13,698 offspring with rare diseases, obtained as part of the UK 100,000 Genomes Project. We identify 1,870 dnSVs, constituting the largest dnSV dataset reported to date. Complex dnSVs (n = 158; 8.4%) emerge as the third most common type of SV, following simple deletions and duplications. We classify 65% of these complex dnSVs into 11 subtypes. Among probands with dnSVs (n = 1,696), 9% exhibit exon-disrupting pathogenic dnSVs associated with the probands' phenotype. Notably, 12% of exon-disrupting pathogenic dnSVs and 22% of de novo deletions or duplications previously identified by array-based or whole-exome sequencing methods are found to be complex dnSVs. We also find distinct genomic properties of de novo deletions depending on the parent of origin. This study highlights the importance of complex dnSVs in the cause of rare disorders and demonstrates the necessity of specific genomic analysis to avoid overlooking these variants.

Structural variants (SVs), defined as genetic changes ≥50 bp that encompass copy number variants (CNVs)[1], rearrangements, and mobile element insertions, play an important role in cancer when occurring in somatic cells[2]. They also arise in the germline, with de novo structural variants (dnSVs) contributing to rare disorders[3–10]. For instance, chromosomal microarray (CMA), which is capable of detecting submicroscopic CNVs, demonstrated an average diagnostic yield of 12.2% in patients with developmental and intellectual disorders[11]. Beyond CNVs, other types of SVs, such as complex SVs involving clustered breakpoints originating from a single event[3,12], have provided insights into the genetic aetiology of rare disorders[13–15], surpassing the explanatory power of CNVs alone.

Nevertheless, in contrast to de novo single-nucleotide variants (SNVs), there is limited information on the prevalence and characteristics of dnSVs, particularly complex rearrangements, in rare disorders, primarily due to the substantial technical challenges associated with their detection[16].

One prominent difficulty arises from the inherent limitations of short-read technologies in accurately capturing and characterising large-scale genomic rearrangements. The restricted read lengths often result in fragmented or incomplete representations of complex structural variations, leading to difficulties in assembling the complete picture of genomic architecture. This issue is particularly pronounced in regions with high sequence similarity, where distinguishing between

[1]Wellcome Sanger Institute, Wellcome Genome Campus, Hinxton, UK. [2]Genomics England, London, UK. [3]Department of Clinical Genetics, Cambridge University Hospitals, Cambridge, UK. [4]Department of Genomic Medicine, University of Cambridge, Cambridge, UK. [5]European Molecular Biology Laboratory, EBI, Hinxton, Cambridge, UK. [6]Department of Genetics, University of Cambridge, Downing Street, Cambridge, UK. ✉e-mail: hj6@sanger.ac.uk; rr11@sanger.ac.uk

homologous sequences presents significant computational and analytical challenges.

Long-read sequencing mitigates the challenges associated with short-read platforms by offering a more direct span across SVs, thereby enabling better resolution and a more complete representation of complex genomic variations. While long-read sequencing offers unique advantages in studying SVs, the lack of substantial long-read sequence datasets from rare disorder cohorts highlights the ongoing importance of precise short-read based SV analytical pipelines[17–19]. These pipelines, essential for detecting a broad spectrum of SVs and reducing false positives. This is particularly pertinent in the absence of large cohort population datasets, which hampers accurate filtering and necessitates robust short-read analytical approaches[20]. Consequently, leveraging large-scale short-read sequence datasets with rigorous analytical approaches remains key for a nuanced understanding of SVs in diseases, particularly rare disorders.

To shed light on their significance in rare diseases, we analysed dnSVs identified in 13,702 whole-genome-sequenced parent–child trios from 12,568 families from the rare disease programme of the 100,000 Genomes Project (Supplementary Data 1)[21] using rigorous analytical approaches. The rare disease cohort encompasses individuals with a broad spectrum of conditions, with neurology and neurodevelopmental (NN) disorders making up half of the cohort. Other represented disorders include ultra-rare conditions, ophthalmological, renal and urinary tract, cardiovascular, endocrine, and additional disease groups (Supplementary Fig. 1a).

## Results

### Rate of de novo SVs and parental age and sex bias

We developed a rigorous pipeline to analyse an average of 13,980 candidate variants (standard deviation = 2550) per proband, already called by Genomic England using the Manta caller[22]. We identified a total of 1870 high-confidence dnSVs (Fig. 1 and Supplementary Data 2), all of which were visually inspected ("Methods"). Some of these dnSVs were validated using previously identified dnSVs detected in independent sequencing data from overlapping family cohorts. The validation rate was 100% (n = 44): 37 candidate dnSVs were confirmed by array/whole-exome sequencing from the Deciphering Developmental Disorders (DDD)[23] study and 7 candidate dnSVs were confirmed using long-read sequencing data from Genomic England (GEL)[21], respectively (Supplementary Fig. 2). In addition, we validated 11 pathogenic dnSVs (Fig. 2 and Supplementary Fig. 3) using RNA-seq data by confirming abnormal RNA reads supporting dnSVs (Supplementary Fig. 3a), underexpression (Fig. 2g), and Supplementary Fig. 3b), or aberrant splicing patterns (Supplementary Fig. 3c). Furthermore, 19 of the pathogenic dnSVs involving inversions were validated by an independent group[24].

Using 1870 high-confidence dnSVs from 1696 probands (91% of probands had a single SV; Supplementary Fig. 1b), we estimated an overall mutation rate of 0.13 events per genome per generation, in line with previous reports[25–27] (Supplementary Fig. 4a). The rate of dnSVs varies across the rare disorder categories; such that probands with NN disorders and those with cardiovascular disorders exhibit the highest dnSV rate (0.15 event per genome), whereas probands with ophthalmological and hearing and ear disorders show the lowest (0.1 event per genome; Supplementary Fig. 4b). It is worth noting that the rate of dnSVs is marginally higher in the probands (0.13 event per genome) than in unaffected siblings (n = 207; 0.09 event per genome; P = 0.05). Approximately 12% (n = 1696) of the probands harboured at least one dnSVs. We identified 4 individuals with a considerably higher number of dnSVs (n ≥ 4). These individuals, recruited under different rare disease categories, are not among the previously reported germline SNV hypermutators[28] and have no known history of parental exposure to chemotherapy. Unlike the known multiple dnCNVs phenomenon that shows a predominance of copy number gain[29,30], 88% of the

identified dnSVs in these individuals were a deletion (median = 1.5 kb), suggesting that further investigation is needed to characterise the multiple dnSVs in these individuals. We found a statistically significant positive correlation between the number of dnSVs and de novo SNVs/indels (P = 3.87E-07; Supplementary Fig. 5a), which is partly explained by the parental age effect[27,31]. However, the mechanistic basis of this correlation remains unclear.

Interestingly, we observed a greater enrichment of dnSVs in probands without diagnostic SNVs/indels compared to those with diagnostic SNVs/indels (P < 5.00E-02; Fig. 1a), suggesting that a significant proportion of unsolved cases is likely to be explained by dnSVs. We also found a parental-age effect on the occurrence of dnSVs (Fig. 1b, $P_{paternal} < 5.00E-02$ and $P_{maternal} < 1.00E-02$). Overall, we identified a significant increase in parental age at birth in probands with dnSVs compared to those without (P < 5.00E-02). Among the rare disorder classes, a significant difference in parental age distribution is observed in dysmorphic and congenital abnormality syndromes and skeletal disorders (P < 5.00E-02), while only the association with skeletal disorders remained significant after multiple testing correction (Benjamini-Hochberg corrected P < 0.05; Supplementary Fig. 5b). Additionally, we observed 67.8% of the phased dnSVs originated from paternal germ cells (Supplementary Fig. 5c), as a proportion consistent with previous studies on structural variation (66–74.4%)[26,27]. This finding aligns with the well-documented paternal bias in de novo SNVs and indels, reinforcing the broader trend of increased germline mutagenesis in the male lineage[32].

### Distribution of different classes of dnSVs

Among the different classes of dnSVs (Fig. 1c), simple deletion (n = 1377; 73.6%) was the most common, followed by tandem duplication (n = 245; 13.1%). The median detected deletion and tandem duplication sizes are 3.7 kb (range 52 bp – 61 Mb) and 49 kb (range 135 bp – 154 Mb), respectively (Supplementary Fig. 5d). Furthermore, we identified other classes such as complex SVs (n = 158; 8.4%), reciprocal inversion (n = 49; 2.6%; Fig. 2a), reciprocal translocation (n = 30; 1.6%; Fig. 2b), and templated insertion (n = 6; 0.7%; Fig. 2c). The representative probands with simple SVs disrupting phenotype-relevant genes are shown in Fig. 2a–c. For example, we identified a templated insertion that disrupted MECP2, which has a well-established function in neurodevelopment[33] in probands with NN disorders. This gene is known to be recurrently affected by dnSVs[10] (Fig. 2c). This was independently validated by long-read sequencing (Supplementary Fig. 2b).

We inferred the timing of maternally derived duplication formation into meiosis I and II based on the fact that heterologous allele duplications are known to occur only before the separation of homologous chromosomes during meiosis I. In contrast, homologous allele duplications are known to occur before the separation of sister chromatids during meiosis II[34]. We identified 41 cases of dnSVs with duplications, which comprise 30 tandem duplications and 11 complex SVs (median = 380 kb; range 38 kb – 40 Mb), for timing analysis. We classified the timing of duplication of maternal origin into meiosis I and II (Methods and Supplementary Fig. 6). This classification revealed that 85% of duplications in this cohort originated from maternal meiosis II (P = 4.87E-06; Fig. 1d). Further investigations using larger cohorts are required to confirm which step of meiosis contributes more significantly to dnSVs[34,35].

### The role of complex SVs in rare disorders

Notably, the third most common type of dnSVs is complex SVs. We further classified complex SVs into nine major classes (Fig. 1e). The most common class, termed 'Loss-Loss' (n = 18;11.4%), comprised two adjacent deletions (Fig. 2d). For instance, the two adjacent deletions (2 kb and 3 kb in length) in a proband with an NN disorder affected two exons in CNOT2 (Fig. 2d) for which haploinsufficiency is known to cause a neurodevelopmental disorder with characteristic facial

features[9]. In addition, adjacent deletions (5 kb and 1.7 kb in length) in a proband disrupted exon 2 of *AMER1* (Supplementary Fig. 2c), for which deficiency is associated with osteopathia striata with cranial sclerosis[36]. Other classes comprising inversion and deletion are 'Inv-Loss' (i.e., inversion with flanking deletion; *n* = 14;8.9%; Fig. 2e) and 'Loss-Inv-Loss' (i.e., paired deletion inversion; *n* = 12; 7.6%; Fig. 2f). The representative cases with these types of complex SVs disrupting renal and urinary tract disorder-[37] and NN disorder-related genes[38], such as *KMT2A, AFF2, FMR1*, and *SRRM2*, are shown in Fig. 2e–g. We observed significantly reduced mRNA expression of *SRRM2* (*P* = 4.00E-04), disrupted by 'Loss-Inv-Loss' in a proband with an NN disorder (Fig. 2g).

Another commonly observed class, termed 'Loss-invDup' (*n* = 14; 8.9%), is characterised by copy-number loss plus a nearby duplication linked by inverted rearrangements. For instance, a 'Loss-invDup' in a proband with NN affected an exon in *AUTS2* (Fig. 3a), which has been

implicated in neurodevelopment and as a candidate pathogenic gene for numerous neurological disorders[39]. Another class, 'Deletion bridge' (i.e., bridge of templated insertion; *n* = 7; Fig. 3b and Supplementary Fig. 7a, b), led to large deletions (15 Mb in chromosome X in Fig. 3b and 720 kb in chromosome 1 in Supplementary Fig. 7b) containing genes involved in neurodevelopment[40,41], such as *GALNT2, MECP2*, and *CTNNB1*, in probands with NN disorders (Supplementary Fig. 7a, b). 'Translocation-Loss' (*n* = 3) led to deletions on either one chromosome (Fig. 3c) or both chromosomes (Supplementary Fig. 7c), resulting in the disruption of phenotype-relevant genes such as *ARID1B*[42] in a proband with an NN disorder.

The other remaining classes, comprising duplication and inversion, are 'DUP-NML-DUP'[3,43] (i.e., paired duplication inversion or Dup-invDup; *n* = 7; 4.4%; Supplementary Fig. 8) and 'DUP-TRP/INV-DUP'[15] (i.e., Dup-Trp-Dup; *n* = 14; 8.9%; Supplementary Figs. 7d and 8),

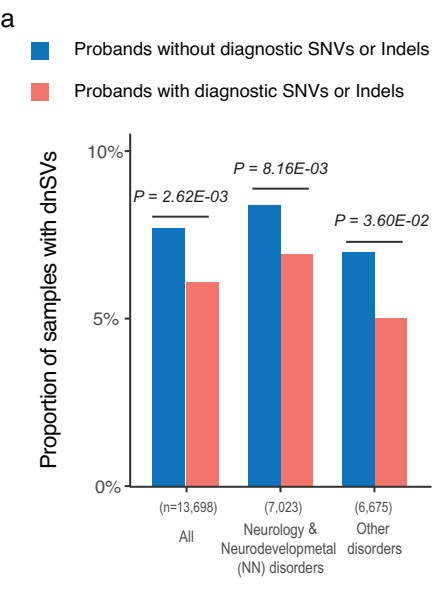

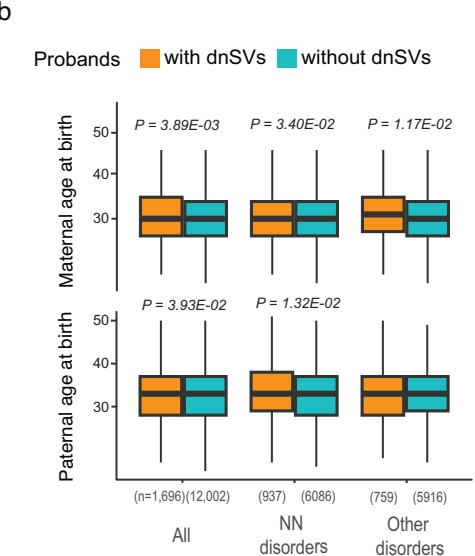

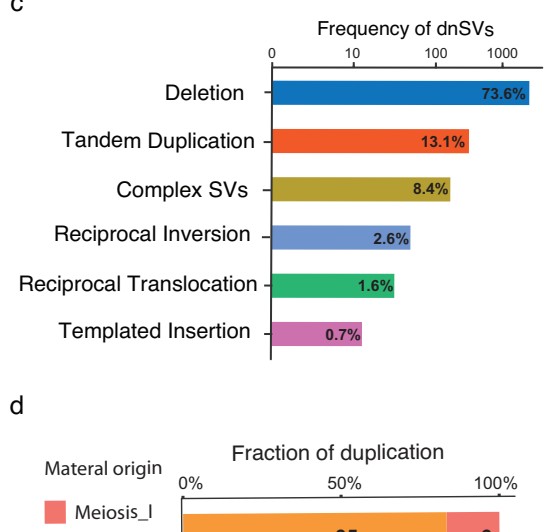

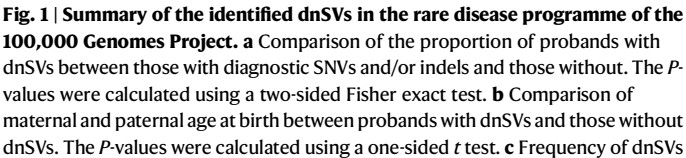

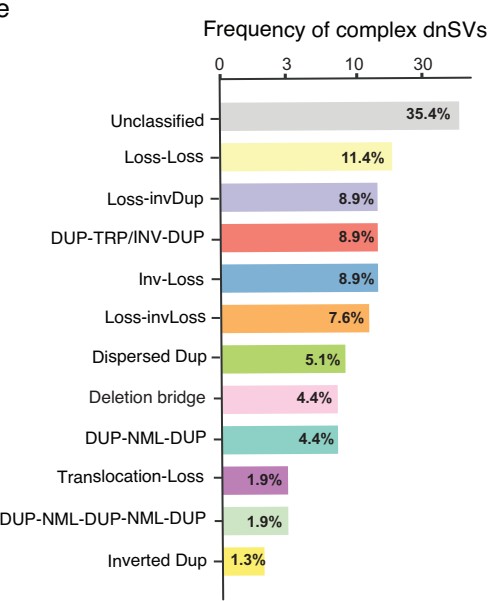

**Fig. 1 | Summary of the identified dnSVs in the rare disease programme of the 100,000 Genomes Project. a** Comparison of the proportion of probands with dnSVs between those with diagnostic SNVs and/or indels and those without. The *P*-values were calculated using a two-sided Fisher exact test. **b** Comparison of maternal and paternal age at birth between probands with dnSVs and those without dnSVs. The *P*-values were calculated using a one-sided *t* test. **c** Frequency of dnSVs

classes. The box plots display the median (centre line) and interquartile range (25th to 75th percentile; boundaries of the box), with whiskers indicating minima and maxima. **d** Timing of duplications from maternal origin. Fraction of duplications according to the timing. The *P*-value was calculated using a two-tailed exact binomial test. **e** Frequency of complex dnSVs classes.

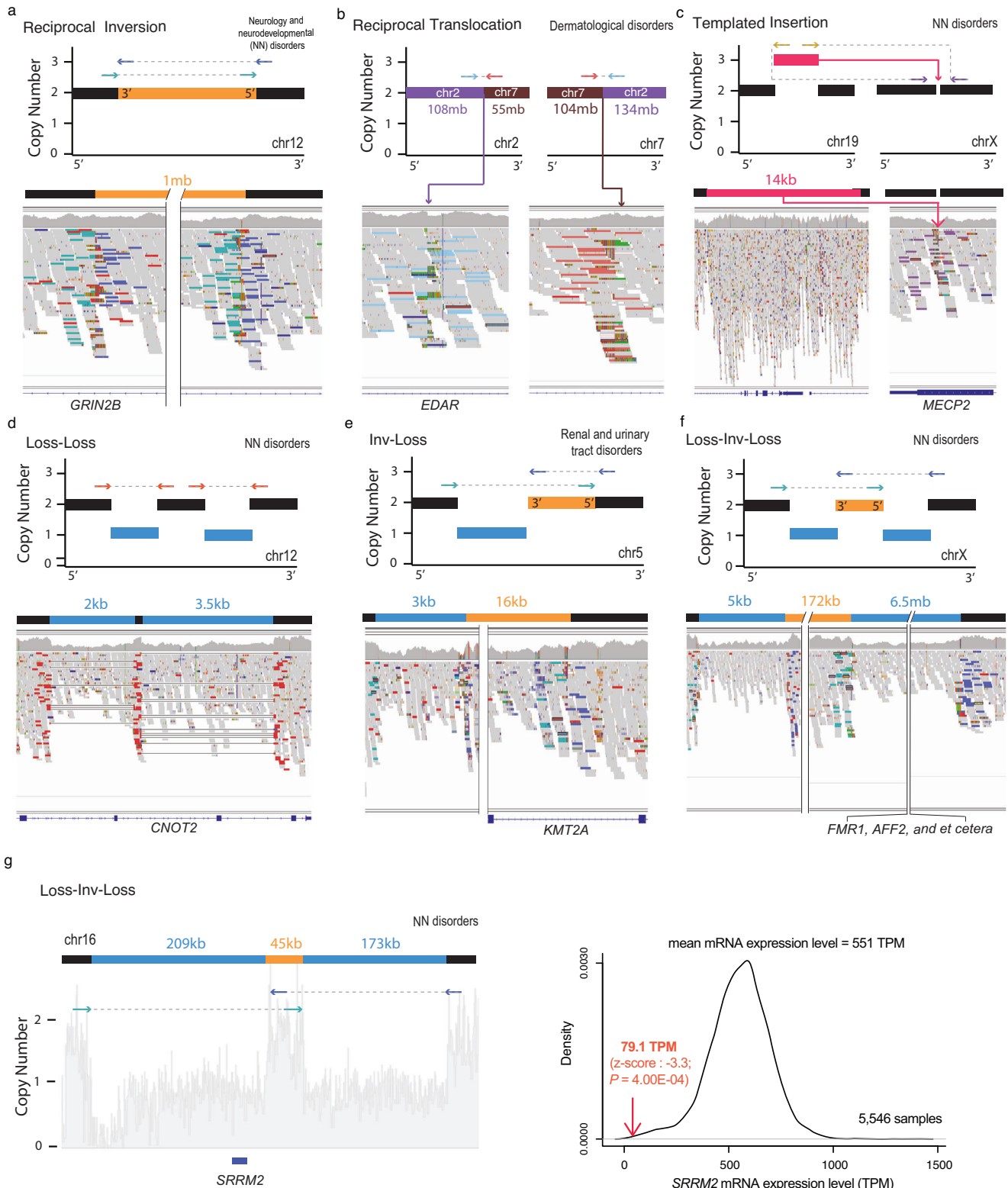

**Fig. 2 | Representative complex SVs disrupting potential causal genes.**
**a**–**c** Simple SVs (**d**–**g**) complex SVs. **a**–**g** Schematic of major dnSV classes with copy numbers and read patterns (top). The schematic segments in blue, red, and orange represent deletion, duplication, and inversion, respectively. The size of the segments is not proportional to the SV size indicated above the segments. Integrative Genomics Viewer (IGV) screenshots illustrate dnSVs in probands (bottom). The potential causal genes affected by dnSVs are marked below the screenshots. **g** Distribution of mRNA expression level of the SRRM2 gene from 5546 samples (right). The expression level from the proband (left) is indicated by the red arrow (right). The P-value was calculated from a one-tailed z-score test.

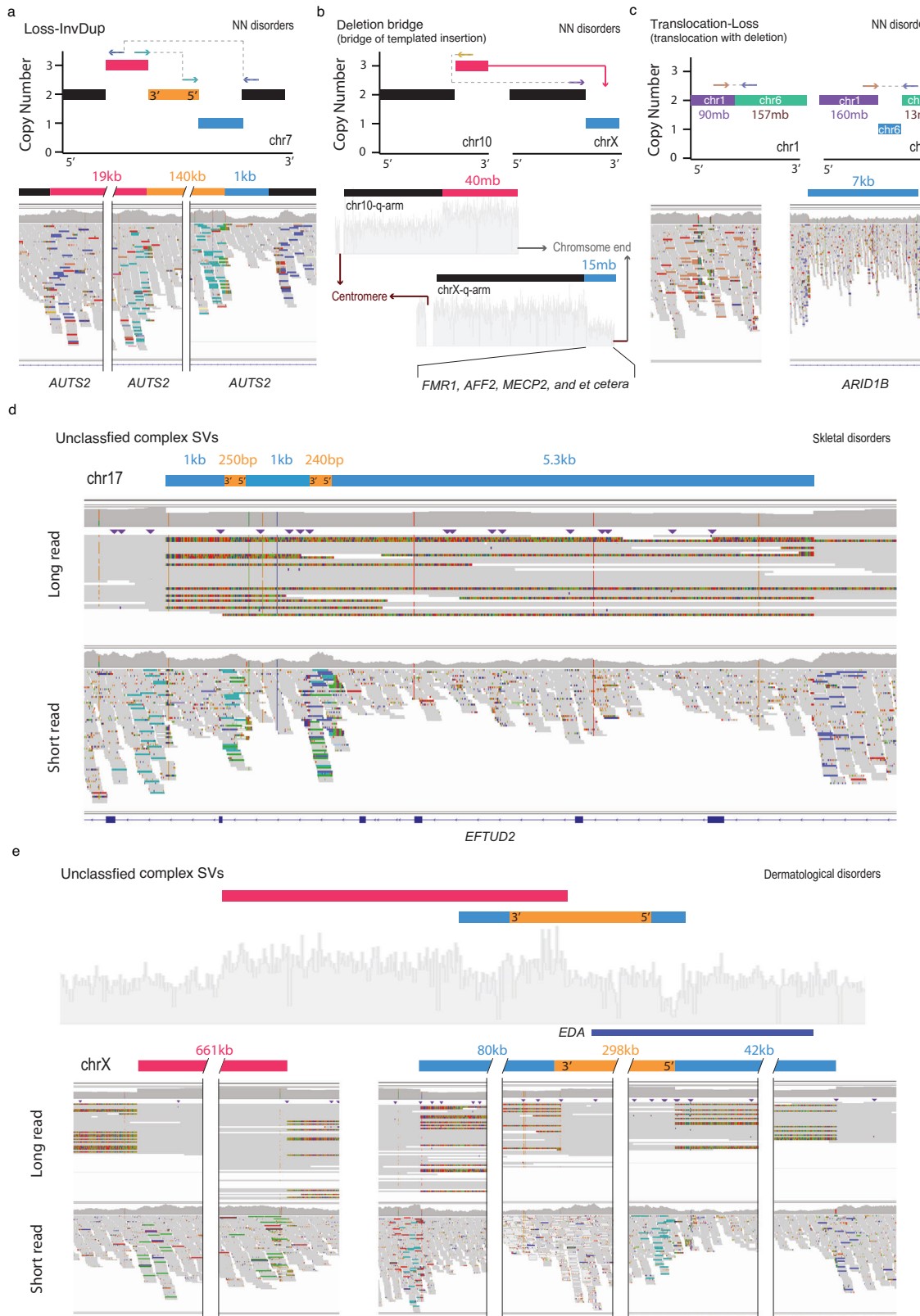

**Fig. 3 | Representative complex SVs disrupting potential causal genes.**
**a**–**c** Classified complex SVs (**d**, **e**) unclassified complex SVs. **a**–**e** Schematic of major dnSV classes with copy numbers and read patterns (top). The schematic segments in blue, red, and orange represent deletion, duplication, and inversion, respectively. The size of the segments is not proportional to the SV size indicated above the segments. IGV screenshots illustrate dnSVs in probands (bottom). The potential causal genes affected by dnSVs are marked below the screenshots. **d**, **e** Unclassified types of complex dnSVs. The schematic segments (**d**, **e**) and coverage plot (**e**) are shown above IGV screenshots displaying long-read (top) and short-read (bottom) in probands.

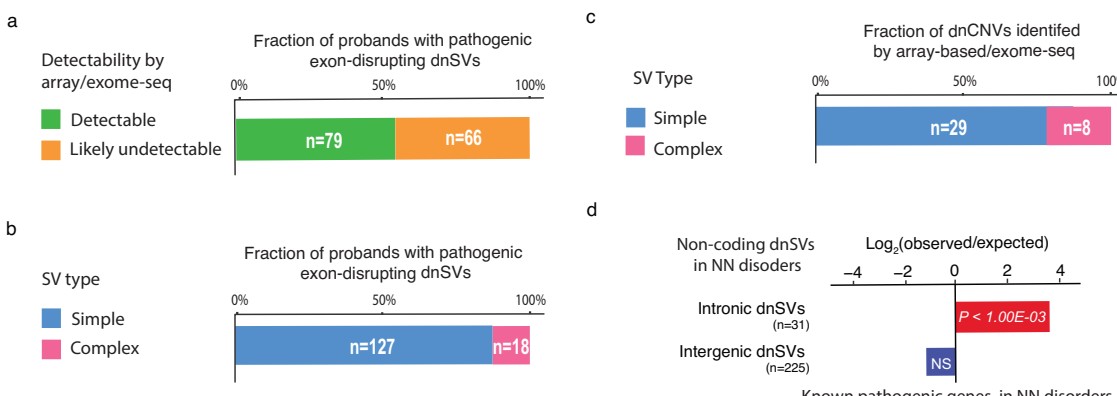

**Fig. 4 | Clinical relevance of dnSVs. a** Percentage of probands with pathogenic exon-disrupting dnSVs according to detectability by array-based / whole-exome sequencing. **b** Percentage of probands with pathogenic exon-disrupting dnSVs by SV type. **c** Percentage of dnCNVs identified by array-based / whole-exome sequencing by SV type. **d** Over-representation of intronic dnSVs in known pathogenic genes in NN disorders. A window of 50 kb up- and downstream was added to each intergenic dnSV for this enrichment analysis. The *P*-value was estimated using a one-sided permutation test.

exhibit structures involving two duplications linked by inverted rearrangements and duplication–inverted-triplication–duplication, respectively. Interestingly, beyond local-2 jumps (i.e., clusters of two rearrangements) as described above, we also found three instances of 'DUP-NML-DUP-NML-DUP'[44–46] (i.e., known as local-3-jumps in the cancer field[2]) involving three local rearrangements (Supplementary Fig. 9a). Furthermore, other complex duplications, such as dispersed ($n = 8$; 5.1%) and inverted duplications ($n = 2$; 1.3%), were also observed. Although most pathogenic effects of these complex SV types involving duplication arise from overexpression of triplo-sensitive genes[15,45] (i.e., gain-of-function), these variant types have been reported to cause disease by loss-of-function mechanisms such as gene disruption[47], gene fusion at breakpoints[48], and segmental uniparental disomy[49].

The complex SVs that did not fit into the described classes were categorised as 'Unclassified' ($n = 56$; 35.4%). Two of the complex SVs under the "Unclassified" category had long-read data that enabled us to resolve their genomic configuration (Fig. 3d, e). A proband with a skeletal disorder (Fig. 3d) had a deletion-inversion-deletion-inversion-deletion structure, which affected several exons in *EFTUD2*, for which deficiency is likely to lead to craniofacial anomalies[50]. Another case has a structure of duplication followed by 'Loss-Inv-Loss' which disrupted the phenotype-relevant gene, *EDA*[51] (Fig. 3e). In addition, a case with a small deletion (3 kb in chr22q.13.33) within a large deletion (20 kb in chr22q.13.33), where one of the breakpoints was the same, in a proband with an NN disorder disrupted *SHANK3* associated with a broad spectrum of neurodevelopmental disorders[52] (Supplementary Fig. 9b). Collectively, these results highlight the complex nature of dnSVs in rare disorders.

## Clinical impact of dnSVs

Overall, our analysis reveals that among probands with dnSVs, 9% (145/1696) exhibit exon-disrupting pathogenic dnSVs associated with the probands' phenotype. Notably, 66 of these 145 (46%) pathogenic SVs were balanced rearrangements (e.g., reciprocal inversion) or CNVs affecting <3 exons that cannot be reliably detected by array-based or whole-exome sequencing methods (Fig. 4a and Supplementary Data 2), highlighting the importance of WGS-based genetic testing in routine clinical care.

In our study, we observed that 1.4% of probands with NN disorders harboured complex dnSVs, a prevalence about two times higher than in a previous autism spectrum disorder study[27] (0.76%; $P = 2.00E-02$; two-sided Fisher exact test). Notably, approximately 12% of pathogenic dnSVs in our dataset were identified as complex events (Fig. 4b),

highlighting their significant role in rare disorders despite their lower frequency. Moreover, among probands with array-based or whole-exome sequencing data available, 22% of de novo CNVs identified by these data (8/37) were complex SVs, which were previously mis-classified as simple dnSVs (Fig. 4c).

Furthermore, our investigation reveals distinctive patterns within intronic and intergenic dnSVs among probands with NN disorders. Intronic dnSVs showed a significant enrichment in known pathogenic genes associated with NN disorders in the G2P database[53] ($P = 1.00E-03$). In contrast, intergenic dnSVs, when assessed for genes within a 50 kb range up- and downstream of the dnSVs ("Methods"), did not show such an association, suggesting the potential pathogenic role of intronic dnSVs in rare disorders (Fig. 4d). Additional studies using RNA-seq and/or CRISPR/Cas-9 genome editing are needed to elucidate the functional impact of these intronic dnSVs on mRNA splicing and expression.

## Genomic properties of dnSVs

In exploring genomic properties of dnSVs, we observed a prevalent distribution of de novo deletions (dnDELs) and tandem duplications (dnTDs) in gene-dense areas (Supplementary Fig. 10), in line with previous findings in somatic cells[2]. However, smaller dnDELs (< 10 kb) are enriched in early-replicating regions ($P = 1.00E-03$; Supplementary Fig. 10a), which is inconsistent with previous reports[2].

We observed that the majority of dnSVs, primarily simple dnDELs, exhibit enrichment at the subtelomeric regions across autosomes (Supplementary Fig. 11). We also identified a positive association between the number of subtelomeric dnDELs and early replication regions, especially when they were within 15 Mb of telomere ends (Spearman's rho = 0.56, $P = 6.46E-03$; Supplementary Fig. 12). In total, the density of dnDELs within 15 Mb of telomere ends (i.e., 1.3/Mb) is 2.8 times greater than the autosome-wide average (i.e., 0.457/Mb).

We observed a distinctive sex difference in patterns of dnSVs, specifically, maternal dnSVs were enriched for larger deletions, while paternal dnSVs were enriched for smaller deletions ($P = 4.99E-05$; Fig. 5a). We further confirmed a similar enrichment pattern using an independent dataset[27] ($P = 1.63E-03$; Fig. 5a). This gender-specific difference is potentially in line with a higher incidence of aneuploidy in oocytes than in sperm[54]. The higher rate of aneuploidy is known to be associated with the distinct features of oocytes[55], such as the architecture of the meiotic spindle, the level of cortical tension at the oocyte surface, weaknesses in surveillance mechanisms that monitor chromosome segregation, and environmental factors. Additionally, we

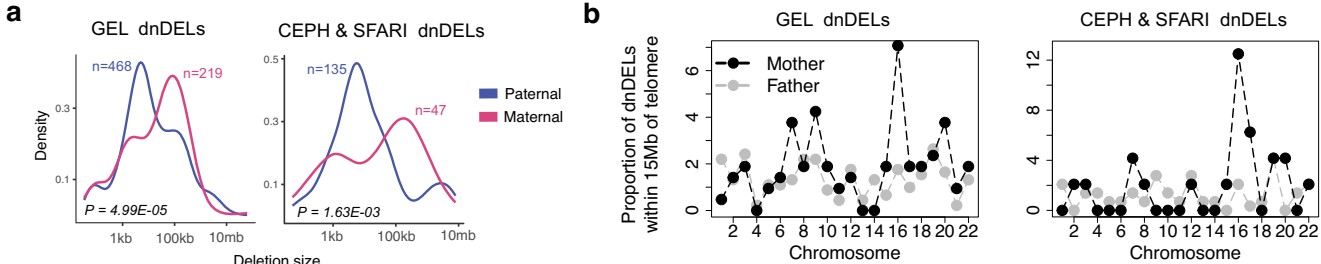

**Fig. 5 | Size and genomic distribution of dnDELs according to parent of origin.**
**a** Comparison of the size of dnDELs according to parent of origin in the GEL cohort (left) and CEPH & SFARI cohort (right). The *P*-values were calculated using a two-sample Kolmogorov-Smirnov test. **b** Proportion of dnDELs within 15 Mb of telomere ends according to parent of origin in the GEL (left) and in the CEPH & SFARI cohort (right).

found that maternal dnDELs are enriched at the subtelomeric regions (< 15 Mb) of chromosome 16 (Fig. 5b). We observed similar maternal enrichment of dnDELs in subtelomeric regions of chromosome 16 in an independent cohort[27] (Fig. 5b), suggesting possible sex-specific mechanisms in the generation of dnSVs.

## Discussion

Our investigation provides a substantial advancement in the understanding of de novo structural variants in rare disorders, encompassing an extensive cohort of 13,702 parent–child trios. In particular, we provide insights into the role of complex SVs in the aetiology of rare disorders.

The prevalence of dnSVs, affecting 12% of probands, highlights the importance of integrating these variants into the broader spectrum of genetic factors contributing to rare disorders. Unlike conventional cytogenetic methods, such as array Comparative Genomic Hybridisation (CGH)-based technologies, WGS offers unparalleled precision in characterising the genomic configuration of complex dnSVs. This is particularly crucial as some simple deletions and insertions may be integral components of complex SVs often overlooked by array-based / whole-exome sequencing methods. For instance, in 37 cases where array and or whole-exome sequencing data were available, we found that 8 complex dnSVs (22%) were misclassified as simple dnSVs. In addition, 66 of 145 (46%) pathogenic SVs identified in our study were balanced rearrangements (e.g., balanced inversion) or CNVs affecting ≤ 2 exons that can't be detected by array-based / whole-exome sequencing methods, highlighting the importance of WGS-based genetic testing in routine clinical care.

Notably, dnSVs exhibited non-random distribution patterns, showing enrichment in specific genomic locations associated with distinct features depending on the parent of origin. Strikingly, we observed an enrichment of maternal dnDELs within 15 Mb of the telomeres of chromosome 16. This enrichment positively correlates with skewed early replication regions across chromosomes. While the genomic basis for this maternal bias in subtelomeric regions of chromosome 16 is unknown, previous reports have suggested potential explanations[56]. These include early subtelomeric replication in meiosis[57], increased rates of meiotic double-strand breaks in the distal parts of chromosomes[58], or biased maternal non-crossover gene conversion[59]. Overall, these findings indicate the need for further investigation into parental influence and region-specific impacts on disease manifestation.

We note several limitations in our approach, which open potential avenues for future investigations. Complex structural variants are still underestimated in our cohort because of the inherent limitations of short-read-based SV discovery. For example, SVs in repeat-rich regions (e.g., segmental duplications or retrotransposons) remain challenging to identify. Although we identified some of these SVs using read-depth-based algorithms (e.g., CANVAS), resolving the genomic configuration of complex SVs using read-depth-specific calls that do not provide read orientation, along with read-pattern-based calls, is challenging. Furthermore, short-read sequencing is known to fail to capture large SVs, especially large insertions. Due to the lower detection sensitivity of de novo retrotransposition, with the current pipeline, these variants have been excluded from our analysis. Specifically, we observed a rate of 0.01 events per genome, which is far lower than expected (0.03-0.038 per genome[27,60]). Future research could explore techniques such as long-read sequencing[61] to enhance our ability to detect dnSVs in repetitive regions[62]. Finally, the systematic identification of gene duplication leading to triplosensitivity[63] and inherited pathogenic SVs will be needed to facilitate more comprehensive diagnostics.

Overall, our findings expand the understanding of dnSVs in rare disorders and highlight the need for ongoing research to unravel the complexities of their contribution to the aetiology of rare disorders and the potential for clinical application.

## Methods
### SV calling
We used available data from the rare disease cohort of the 100,000 Genomes Project generated by Genomics England[21]. The 100,000 Genomes Project was approved by the East of England—Cambridge Central Research Ethics Committee (REF 20/EE/0035). Genomic blood DNA libraries were prepared using the Illumina TruSeq PCR-free protocol, and whole-genome sequenced on the Illumina HiSeq X platform (2 × 150-bp paired-end reads). Read alignment and SV calling using Isaac[64] and Manta[22], respectively, were performed by the Genomics England Bioinformatics team[21]. The details of sequencing and variant calling have been previously described[21]. Manta VCF files were converted to BEDPE format using SVtools (v0.5.1)[65] and then BEDTools (v2.31.0)[66] was utilised to extract proband-specific SVs with ≥ 50 bp in length for each family (Supplementary Fig. 13). We first removed SVs on Y chromosome and further removed SVs with evidence of clipped reads (i.e., split reads) at breakpoints in either parent. Specifically, SVs supported by ≥ 4 clipped reads at either breakpoint or ≥ 2 clipped reads at both breakpoints in either parent were excluded. SVs found in 3 or more samples were removed because such SVs were likely alignment artefacts. We selected SVs flagged as "PASS" or "MGE10kb" (i.e., Manta calls with length < 10 kb) and further narrowed down SVs with the Manta score > 30 and supporting discordant reads > 10. We rescued SVs with imprecise breakpoints if they were supported by CANVAS[67]. In detail, SVs tagged as "IMPRECISE" (i.e., imprecise breakpoints) were rescued if they were also flagged as "ColocalizedCanvas". We excluded SVs with VAF < 0.1 (*n* = 10) to remove mosaic SVs. Translocation through retrotransposon-mediated 3′ transduction was excluded to focus on dnSVs. All SVs were manually validated to identify high-confidence de novo SVs using the IGV browser[68]. Specifically, we visually validated the presence of abnormal reads (i.e., discordant/split reads) in probands and the absence of such reads in their parents. We additionally called CNVs (i.e., deletion and duplication)

with > 10 kb using CANVAS to identify more diagnostic CNVs. Read-depth-specific calls were only used to assess their pathogenicity because clustering and classification (i.e., resolving genomic configuration of complex SVs) of read-depth calls that did not have read-orientation were challenging. Long-read sequencing data (23 resequenced samples with PacBio technology), RNA-seq (whole blood RNA sequencing from 5,546 samples) and diagnostic SNV/Indels were obtained from Genomic England[21]. Previous dnCNV calls from array-based or whole-exome sequencing were obtained from the DDD study cohort[23]. Insertion events called by Manta[22] were further classified into retrotranspositions using RepeatMasker. This process resulted in a lower sensitivity because Manta with Issac-based alignment is not optimised to call retrotranspositions. Retrotransposition-specific identification tools, such as MELT[69] or xTea[70], are needed to increase sensitivity for retrotransposition detection. To estimate the proportion of probands with complex SVs in the previous study[27], we obtained 19 probands with complex SVs where "sv_type" was "CPX" and "role" was "proband".

## SV classification

We used ClusterSV[2] (https://github.com/cancerit/ClusterSV) to group rearrangements (i.e., breakpoints), into rearrangement clusters. The key advantage of ClusterSV is to identify clusters of dispersed breakpoints[46] without requiring a predefined distance threshold, allowing for a data-driven detection of complex genomic rearrangements. We defined complex SVs as those with ≥ 2 clustered breakpoints except for simple SVs involving reciprocal inversion, balanced translocation, templated insertion, and dispersed duplication. In general, we classified the types of complex SVs according to the previous study that comprehensively characterised somatic complex SVs using thousands of cancer genomes[2]. In short, complex SVs involving two inversions were categorised into Loss-invDup, DUP-TRP/INV-DUP, Inv-LossDU (i.e., inversion with flanking deletion), Loss-invLoss (i.e., paired deletion inversion), and DUP-NML-DUP (i.e., paired duplication inversion) according to read patterns and copy numbers (Supplementary Fig. 8). Complex SVs involving two deletions were classified as Loss-Loss. Bridge deletion (i.e., bridge of templated insertion) and Translocation-Loss (i.e., translocation with deletion) were classified using the previously described criteria[2,71]. DUP-NML-DUP-NML-DUP (i.e., Local-3 jumps) involving three local rearrangements were discovered according to the read patterns described in the previous cancer study[2]. Breakpoints filtered out near unresolved SV classes were rescued if they could resolve the configuration of unresolved SV classes according to the types of SV defined. For the remaining unresolved SV, CANVAS calls were used to resolve their genomic configuration manually. Complex SVs that did not fit into the described classes were categorised as 'Unclassified'. All complex SVs were manually validated using IGV browser (2.18.2)[68], Samplot (v1.3.0)[72], or BamSnap (v0.2.19)[73].

## SV phasing to identify parent of origin and estimation of the timing of duplication from maternal origin

We used unfazed (v1.0.2)[74], which employs both extended read-backed- and SNV allele-balance- phasing, to identify the parent of origin for dnSVs. Haplotypecaller (v3.3)[75] was utilised to make an input for unfazed. Phasable dnSVs (51%; 962/1870) were used for downstream analysis concerning the parent of origin. To classify the timing of maternally derived duplication into meiosis I and II, we first identified duplication (including those in complex SVs) from maternal origin (step 1) and further classified them into meiosis I and II (step 2) using a set of informative genotypes (Supplementary Fig. 6) For binary classification at each step, the ratio of the number of SNPs supporting one class to another class was calculated, and a class for which the ratio was greater than 0.9 was chosen, At least three SNPs were required for either class at each step. These filtering criteria could time large

duplications with a handful of erroneously called SNPs and remove ambiguous duplications such as those originating from both parents during early development (e.g., potentially due to mitotic crossing-over). The timing of paternally derived duplications was not inferred because duplications can also occur in a premeiotic state during male gametogenesis throughout life.

## Evaluation of clinical relevance of dnSVs

The identified SVs disrupting exons were reviewed for potential clinical relevance by NHS clinical scientists and/or Genomics England. We considered SVs as being potential (likely) pathogenic SVs if at least one of the following criteria were fulfilled: (i) the variant had been clinically assessed as likely pathogenic or pathogenic by an NHS genomic laboratory hub. In detail, the variant had been assessed by an NHS clinical laboratory according to the best practice guidelines recommended by the Association for Clinical Genomic Science (ACGS) as being likely pathogenic or pathogenic and related to the primary phenotype for which the participant was recruited to the 100,000 Genomes Project. (ii) the variant had been reviewed on a research basis and considered to be a strong candidate diagnostic variant. In the research review, variants resulting in loss of function for genes in which haploinsufficiency is a known mechanism of disease or variants resulting in loss of a critical domain impacting genes associated with a phenotype relevant to the primary clinical indication were considered as candidate diagnostic variants.

We classified pathogenic SVs annotated with "reciprocal inversion" or "reciprocal translocation" as not likely to be detected by array-based or whole-exome sequencing methods. In addition, the SVs involving deletions spanning < 3 exons of GENCODE canonical transcript[76] were classified into this category.

## Enrichment testing of non-coding dnSVs in known pathogenic genes in NN disorders

We first extracted the non-coding dnSVs (i.e., intronic and intergenic dnSVs) for which genomic coordinates did not include any exons in NN disorders based on the Gencode basic V45 GTF file and then obtained the known pathogenic genes associated with NN disorders from the Gene2Phenotype developmental disorders panel[53]. Specifically, we kept all genes with organ specificity equal to "Brain/Cognition", allelic requirement equal to "monoallelic_autosomal", and a confidence category equal to "strong" or "definitive" ($n = 190$ genes). We then computed the observed-over-expected ratio for the overlap between the non-coding dnSVs and known pathogenic genes in NN disorders using the Genome Association Tester software[77]. Intronic and intergenic regions were obtained based on the canonical transcript of protein-coding genes in the Gencode basic V45 GTF file using BioMart and GencoDymo R packages, and bedtools[66]. These two regions were used as a workspace in GAT to test the over-representation of dnSVs in intronic and/or intergenic regions of known pathogenic genes. We added a window of 5–500 kb (5, 10, 25, 50, and 500 kb) up- and downstream to each intergenic dnSVs to perform the enrichment test. The number of random samples ("--num-samples") for each GAT run was set to 1000.

## Enrichments near telomeres and centromeres

We equally partitioned the genome into 5 Mb bins based on their distance to the telomere ends. For comparison, we also partitioned the genome based on their distance to the centromeres. For the validation cohort, we downloaded the all_dnsv.csv file from Belyeu et al.[27]. In total, there are $n = 309$ CEPH and SFAI dnSVs across autosomes after removing chrX and chrY. Finally, only $n = 192$ dnDELs were used in the validation analysis.

## Reporting summary

Further information on research design is available in the Nature Portfolio Reporting Summary linked to this article.

## Data availability

The data used in this study can be accessed via the Genomics England Research Environment, a secure cloud workspace. The raw data, including patient profiles and corresponding genomic sequencing data, are only available under restricted access for patient privacy reasons. Access can be obtained by first applying to become a member of either the Genomics England Research Network (https://www.genomicsengland.co.uk/research/academic) or the Discovery Forum (industry partners https://www.genomicsengland.co.uk/research/research environment). The process for joining the network is described at https://www.genomicsengland.co.uk/research/academic/join-gecip.

## Code availability

Custom Python and R scripts for data analysis can be found at https://github.com/hj6-sanger/GEL_SV (version 1.0.0, https://doi.org/10.5281/zenodo.17093113).

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

## Acknowledgements

We thank the families and their clinicians for their participation and engagement, and our colleagues who assisted in the generation and processing of data. We would like to thank Ana Lisa Taylor Tavares for helpful discussions and advice. This research was made possible through access to the data and findings generated by the 100,000 Genomes Project. The 100,000 Genomes Project is managed by Genomics England Limited (a wholly owned company of the Department of Health and Social Care). The 100,000 Genomes Project is funded by the National Institute for Health Research and NHS England. The Wellcome Trust, Cancer Research UK, and the Medical Research Council have also funded research infrastructure. The 100,000 Genomes Project uses data provided by patients and collected by the National Health Service as part of their care and support. This research was supported in part by a Wellcome Trust grant. R.R. was supported by RCUK | Medical Research Council (MRC) (MR/W025353/1) and Cancer Research UK (CRUK).

## Author contributions

R.R. conceived the project. R.R., P.C., and M.H. supervised the project. H.J., T.Y., and R.R. wrote the manuscript; all authors reviewed and edited the manuscript. H.J. and T.Y. led the analysis of the data with help from J.C., P.D., I.G.S., M.D.C.N., and S.W. reviewed the pathogenic complex dnSV candidates. H.F., helped with clinical interpretation of the dnSVs. A.S., M.H., P.C., H.J., and R.R. helped with data interpretation and statistical analysis.

## Competing interests

P.J.C. is a co-founder, shareholder, and consultant for Quotient Therapeutics, M.E.H. is a co-founder of, consultant to and holds shares in Congenica, a genetics diagnostic company. All remaining authors declare no competing interests.
