## [Transparent Peer Review file · Nature Communications]

Complex de novo structural variants are an underestimated cause of rare disorders

Corresponding Author: Dr Raheleh Rahbari

Version 0:

Reviewer comments:

Reviewer #1

(Remarks to the Author)

Jung and colleagues in "Complex de novo structural variants are an underestimated cause of rare disorders" investigate the impact of complex de novo structural variants in rare disorders. They have focused on short-read genome sequencing performed in families participating in the 100,000 Genomes Project with about 13,698 offspring, a huge cohort in rare diseases. Surprisingly, the third most common type of de novo SV, following simple deletions and duplications, are complex SVs (8.4%). This paper puts forward a strong argument for why we should focus on analyzing them. Doing so would help increase the diagnostic yield, especially when CMA has a low yield of 12.2%. It cannot be understated that this paper shows that there is a significant enrichment for dnSVs for probands without a SNV diagnosis. Furthermore, this paper shows that 34% of the pathogenic SVs cannot be detected by array-based/ES. This paper also provides the community with a new pipeline for detecting and filtering dnSVs. Moreover, 22% of the de novo copy-number variants once thought to be simple are complex instead. This is consistent with the observations in Schuy et al. 2022 Trends Genet Nov;38(11):1134-1146 who analyzed specific genomic syndromes, some of which show an even higher number of complex rearrangements in specific disease loci.

This is an important manuscript from one of the largest and well-characterized rare disease projects currently ongoing. Said that, the manuscript seems to have been written in a hurry without providing key information regarding methods and results to allow other groups to repeat similar analysis. We are left wondering whether the complex SVs were resolved or not, what are the limitations of the study, how many estimated complex SVs may explain the phenotype and what were the criteria for that. Moreover, the analysis was performed using concepts developed in the cancer field which may or may not represent findings in rare diseases. In all, the manuscript to be able to communicate with the rare disease field and therefore it will benefit from a revision to clarify concepts, analysis, etc. If that is done it will provide exciting findings for the community and will help to move the field forward.

General comment

The authors are using a terminology that is distinct different from the one used in the rare disease field, including the definition of complex genomic rearrangements (for example of CGR definition see Schuy et al. Trends Genet Nov;38(11):1134-1146 and Liu et al. Curr Opin Genet Dev 2012 Jun;22(3):211-20) which may be acknowledge. The comparison between the terms would be helpful and help understanding of the important limitations of this study. Moreover, the analysis here focuses on structural variants only and does not access copy-number variation which is a key limitation to access pathogenicity in addition to limit validating the structure.

Methods

1-There is no description of the cohort. Please clarify how many probands were included, how many trios, siblings, how many males, females, etc. Methodology starts with the SV calling but there is no information about the short-reads, coverage, library. In the figures we learnt that long-reads were used but no descriptive information about the methods is provided. How many samples had long-reads in addition to short-reads?

2-Please clarify further how the group evaluated the clinical relevance of dnSVs. The description provided is extremely vague and nonspecific.

3- Please consider adding a workflow for the pipeline and filtering process used to identify the dnSVs.

3- CNVs constitute a very important part of the majority of CGRs in rare diseases. In fact, most of the clinical phenotype will arise from haploinsufficient or triplosensitive loci. It seems that only SVs disrupting exons were reviewed for potential clinical relevance which will miss a lot of potentially pathogenic SVs.

4- Please add more information about the pathogenic SVs not detected by array or exome. A table with the type of rearrangement, sizes, genes affected would be helpful.

4- Why were the coordinates converted from hg38 rather than the BAM remapped to hg38? I can see the advantage with the remapping, but it is not clear what is the advantage of the liftover. In fact, I am wondering whether that could add inconsistencies to the mapping and analysis, especially for CGRs.

5-Please clarify what it means to have the SVs “manually” validated. I have not seen alignment of the breakpoint junctions which is usually the analysis that can be performed using split reads from IGV to confirm: 1. the structure of the CGR; 2. Whether the CGR was fully resolved. If junctions are still open ended, that means the structure was not resolved. Relevant for that is the fact that we often identify real pathogenic SVs that have no split reads although they have discordant read pairs. This happens when the breakpoint junction maps to repeats (SINEs, LINEs, etc). In those cases, Sanger or Target ONT (MinION works well), need to be used to resolve the structure. There was no comment about those type of SVs, were they filtered out? In addition, CGRs often have insertions that are larger than the short-reads and can not be resolved without long-reads or long-read PCR. Please clarify how the analysis was performed and data filtered regarding those issues.

6- In the “SV calling” session is written that “SVs found in 3+ samples were removed because such SVs were likely to be germline SVs”. This may be an incorrect statement since the analysis is looking for germline SVs.

7- SVs with VAF < 0.1 were excluded as potential mosaicism. However, it may also be excluding real ones in low coverage regions. Have you checked for that?

8- Please define clustered breakpoints and explain the benefit of using ClusterSV in CGRs for rare diseases and how many CGRs were detected by using this program. This is relevant because most of the rare diseases CGRs do not cluster (except for a few rare cases of chromotripsis) and show breakpoints that map far away from the CNV/SV. They are often copied from regions 20 kb or less away, sometimes from distinct chromosomes. Please see discussion from Carvalho et al. 2013. *Nat Genet.* 2013 Nov;45(11):1319-26

9- What are dispersed duplications? In rare diseases we often see duplications separated by copy-number neutral regions (by CNV analysis) which are often overlooked as CGRs because they can be far apart. However, analysis of junctions shows they are connected, and that are part of the same SV event. Are those the same type? Please see reference here for more details on this type of structure: Gu et al. 2015 *Hum Mol Genet.* 2015 Jul 15;24(14):4061-77. doi: 10.1093/hmg/ddv146 and Liu et al. 2017 *Cell.* 2017 Feb 23;168(5):830-842.e7. doi: 10.1016/j.cell.2017.01.037

10- “Local 3-jumps” is stated as not yet observed in rare diseases, I do not think this is correct as from the plots they look like separated CNVs connected via junctions which correspond to the definition of DUP-NML-DUP or DUP-NML-DUP-NML-DUP used in several papers . It seems that the nomenclature for cancer and rare disease CGRs are different which makes it difficult to compare structure, please comment. Beck et al. *Cell.* 2019 Mar 7;176(6):1310-1324.e10. doi: 10.1016/j.cell.2019.01.045, Bahrambeigi et al. *Genome Med.* 2019 Dec 9;11(1):80. doi: 10.1186/s13073-019-0676-0, Carvalho and Lupski, *Nat Rev Genet.* 2016 Apr;17(4):224-38

11- Line 445: I suggest the author organize and document the code. Please explain in the README what this pipeline does, what are the expected inputs/outputs. Please also make this command-line executable so that users can use this tool easily.

Results

1- It will be helpful to see the size distribution of all SVs identified in this cohort. Is there a bias towards identifying smaller SVs? It looks like deletions are mostly small.

2- Please consider adding the distribution of calls per proband across the cohort (maybe a boxplot)

3- Line 326: very stringent filtering, going to miss recurrent CNVs, even if it is de novo. It is not totally clear why read-depth was not used to check for CNVs in this cohort. Agree that small SVs (smaller than 10 kb) are hard to identify and would call a lot of false-positive CNVs, but they can still be confirmed once identified by Manta. Larger SVs should certainly be checked.

4- Please provide a table with the SVs that were classified as pathogenic/potentially pathogenic (criteria) or VUS, type, size, coordinate, and detailed information about genes affected by the SVs, disease, phenotype of patient. I would suggest a table with the pathogenic CGRs as part of the main text.

5- Was only 2.4% (44 out of 1872) of the SV calls validated by an orthogonal technology? If a SV affects a dosage sensitive gene potentially contributing to the phenotype, was it validated by array, exome, or ddPCR? If disrupting a dosage sensitive gene, was it validated by Sanger or RNA expression? Please clarify.

6- In rare diseases, segmental duplications are often involved in the formation of de novo SVs, CNVs, and CGRs. But they will be missed by Manta and any other variant caller that do not use read-depth. Alus and LINEs are missed sometimes

often too, especially if Alu-Alu or LINE-LINE forming a fusion repeat. Those are all limitations of the approach used here and should be acknowledged. CGRs may be still underestimated in this cohort.

7- Triplications are very important CGRs in rare diseases, so it is really exciting that the group found 41. They do deserve a bit more description and discussion. Are the 41 triplications of the same type? If they are all DUP-TRP/INV-DUP, what are the sizes of the segments? Please provide results to exemplify. Importantly, depending on the structure and gene content it will affect expression in distinct ways. It can cause a more severe phenotype due to higher expression, disrupt a gene or lead to fusion genes (detailed discussions here Grochowski et al. 2024 Cell Genom. 2024 Jul 10;4(7):100590. doi: 10.1016/j.xgen.2024.100590). They can also cause other alterations depending on the mechanism such as imprinting disease (Carvalho et al. Genome Med. 2019 Apr 23;11(1):25).

8- Regarding the mechanism of triplication formation, it is not clear why it was assumed to be a meiotic event upfront. It seems that the analysis is basically phasing of the SNVs, is that correct? If yes, the only information obtained is the origin of the rearranged variant. There are no results provided (for example, how many SNVs per duplication/triplication are informative?) Moreover, triplications were shown to occur in mitosis (paternal germline Carvalho et al. 2011 Nat Genet. 2011 Oct 2;43(11):1074-81. doi: 10.1038/ng.944) or during development with mixed parental origin for TRP and DUP (Carvalho et al. 2015 and references within Am J Hum Genet. 2015 Apr 2;96(4):555-64. doi: 10.1016/j.ajhg.2015.01.021; Liu et al. 2017 Cell. 2017 Feb 23;168(5):830-842.e7. doi: 10.1016/j.cell.2017.01.037.)

9- The finding of multiple dnSVs is very interesting. Are these samples considered to be outliers? Would that finding validated or could be a technical artifact? If confirmed, please consider showing those results as a table with type, sizes and origin of the ancestral chromosome if available. Are they independent events (i.e., the junctions are not connected forming single SV originated in the same event?)? For comparison and mechanistic discussions, multiple CNVs are extremely rare but they offer a glimpse of potential prone to error DNA repair mechanism leading also to higher frequency of SNVs: Liu et al. Cell. 2017 Feb 23;168(5):830-842.e7. doi: 10.1016/j.cell.2017.01.037; Du et al. Genome Med. 2022 Oct 27;14(1):122; Beck et al. Cell. 2019 Mar 7;176(6):1310-1324.e10.)

Minor comments

- 1- Line 261: should probably also mention that chrY is excluded in this study
- 2- Line 330: what is MGE10kb?
- 3- Line 368: Please check the documentation for the program Unfazed, it should be extended read-backed phasing, not read-based phasing.
- 4- Figure1: b) cannot tell how each pair is significantly different. The medians of the boxplots are visually at the same level... c) I think the x-axis is wrong. How can frequency be 1000? Log-scale?
- 5- Figure3: d) typo "intronic"
- 6- FigureS4: c) this does not add up to ~1800 dnSVs, what happened to the rest?
- 7- FigureS5: what is R?V? (I assume these are alleles?)
- 8- FigureS7: should probably caveat that the shown derivative structures are not the only possibilities. Eg dup-trp-dup can have other rearrangements. The original cancer paper also shows that.
- 9- FigureS12: can we have the number of calls per step? Want to know the effect of each filtering step

(Remarks on code availability)

We suggest the author organize and document the code. Please explain in the README what this pipeline does, what are the expected inputs/outputs. Please also make this command-line executable so that users can use this tool easily.

Reviewer #2

(Remarks to the Author)

The study performed by Jung et al. investigates the features of de novo structural variants (dnSVs) in a large cohort of over 13,000 offspring with rare diseases, providing a unique and substantial contribution to the field of genomics and rare disease research.

I was particularly impressed by several aspects of the study:

1. Unprecedented Dataset:

The study utilized the dnSV dataset from over 13,000 offspring in the UK Biobank, the largest of its kind to date. This extensive dataset not only enhances the statistical power of the study but also allows for a more comprehensive exploration of the properties and prevalence of dnSVs. Such large-scale analysis is crucial for uncovering subtle yet significant patterns that smaller studies might miss.

2. Insight into Complex dnSVs:

The study offers new insights into the complexity and clinical impact of dnSVs, particularly those that are challenging to resolve using traditional techniques, e.g. array-based method or exome sequencing. The authors' approach to visualizing these complex variants is exemplary, making their findings accessible and easily interpretable.

3. Clinical Relevance:

By thoroughly examining the clinical implications of dnSVs, the study provides critical information that could influence future diagnostic strategies for rare diseases. The discussion around the paternal origin of these variants and the nuances of maternal vs. paternal contributions adds depth to our understanding of genetic inheritance and its role in disease.

While the study is robust and impactful, I have a few suggestions regarding the origin analysis and interpretation of the paternal and maternal contributions.

1. Clarification on Paternal Origin Effects:

In line 109-110, the authors reference a previous study by Kong et al., which discusses de novo single nucleotide variants (SNVs), to support their findings on the paternal origin of dnSVs. However, since SNVs and structural variants might have different properties, I recommend that the authors clarify this distinction and provide additional context on how their findings compare to existing knowledge in the field of structural variants.

2. Discussion on Maternal vs. Paternal Origins:

The manuscript could benefit from a more detailed discussion of the differences between maternal and paternal origins of dnSVs, especially regarding triplications/de novo duplication (line 126-129). Addressing whether paternal origin de novo duplications can be inferred to meiosis I or II and discussing the ratio between maternal and paternal contributions could offer additional insights, at least for the duplication. In addition, the use of the term 'triplication' might need some clarification. It appears that the term is used to describe genomic loci where the total copy number is 3, with one normal copy allele (copy number = 1) and a mutated allele with a duplication (copy number = 2). However, "triplication" is also commonly used to refer to an allele with three copies, such as in a duplication–inverted-triplication–duplication structure (line 164). Clarifying this terminology in the manuscript could help prevent potential confusion for readers.

3. The distinct sex difference observed in the dnSV pattern is quite intriguing and adds an important dimension to the study. However, the derivation of the 10 kb size cutoff is not fully explained in the text (line 221-223). It would be helpful to provide more detail on how this cutoff was determined. Additionally, have you considered performing an enrichment test with different size cutoffs? Exploring this could yield further insights. Furthermore, the potential mechanisms behind these sex-specific patterns in dnSV generation are of great interest. If you have any hypotheses or proposed mechanisms, including them would greatly enrich the discussion. Finally, sharing the genomic positions of the dnSVs could be very beneficial for the scientific community, as it would facilitate independent validation in follow-up studies.

Minor Revisions:

- a. A typo was noted in Figure 3d ("Inronic" should be "Intronic").
- b. The significance marker in Figure S4b appears to be misaligned.
- c. The term "read-based" in line 265 should likely be "read-depth."

Code review:

- a. Code block 1-510 is written in python syntax, 510-1224 is in R. It would be helpful to separate them into two files with proper file extension, e.g., .py and .R
- b. suggests adding version for bedtools, samplot, and unfazed to help reproduce
- c. some internal scripts are not available for review:
Line 16, 'check.py' is not available
Line 36, 'bed.py' is not available
Line 54, 'stat.py' is not available
- d. Description is missing for internal R functions, e.g., line 540, 578, 614, 676,766,798,848

In summary, this manuscript presents a significant advancement in our understanding of dnSVs and their clinical implications. With minor revisions, I believe this study will make a valuable addition to the literature in genomics and rare disease research.

(Remarks on code availability)

- a. Code block 1-510 is written in python syntax, 510-1224 is in R. It would be helpful to separate them into two files with proper file extension, e.g., .py and .R
- b. suggests adding version for bedtools, samplot, and unfazed to help reproduce
- c. some internal scripts are not available for review:
Line 16, 'check.py' is not available
Line 36, 'bed.py' is not available
Line 54, 'stat.py' is not available
- d. Description is missing for internal R functions, e.g., line 540, 578, 614, 676,766,798,848

Reviewer #3

(Remarks to the Author)

(Remarks on code availability)

Version 1:

Reviewer comments:

Reviewer #1

(Remarks to the Author)

These are excellent response letter and revised manuscript drafts. The authors addressed all of my previous concerns and clarified remaining questions. This is a beautiful and highly relevant manuscript for the rare disease community.

I have two additional minor comments:

1- There are several spelling errors in figures; please review them carefully. For example in Figure 1: reciprocal, Maternal, Translocation, etc. Also, there are instances where the word "triplication" is used but likely meant to be "duplication" (eg.,

statements in lines 139 and 399), please double-check

2-The authors should consider adding information about the number of the potential clinically relevant dnSVs in the abstract.

(Remarks on code availability)

Reviewer #2

(Remarks to the Author)

All my comments have been addressed.

(Remarks on code availability)

Reviewer #3

(Remarks to the Author)

(Remarks on code availability)

RESPONSE TO REVIEWERS' COMMENTS

We appreciate the time and thoughtful feedback provided by the reviewers on our manuscript, “Complex *de novo* structural variants are an underestimated cause of rare disorders”. Their comments have been invaluable in refining and strengthening our work, and we are pleased to submit a revised version along with a detailed point-by-point response addressing their concerns. We would also like to apologise for the delay in resubmission. The additional time was necessary due to several technical and logistical challenges, including coordination with clinical collaborators to validate and interpret key events; delays in gaining approval to export RNA data from the GEL environment; temporary issues with access to the GEL research platform; The addition and harmonisation of CNV analyses. The original reviewer’s comments and questions are in black, while our responses are in blue. Line numbers corresponding to changes in the Manuscript and Suppl. Materials are noted throughout and should also be visible as tracked changes in the attached source files.

Reviewer #1 (Remarks to the Author):

Jung and colleagues in “Complex *de novo* structural variants are an underestimated cause of rare disorders” investigate the impact of complex *de novo* structural variants in rare disorders. They have focused on short-read genome sequencing performed in families participating in the 100,000 Genomes Project with about 13,698 offspring, a huge cohort in rare diseases. Surprisingly, the third most common type of *de novo* SV, following simple deletions and duplications, are complex SVs (8.4%). This paper puts forward a strong argument for why we should focus on analyzing them. Doing so would help increase the diagnostic yield, especially when CMA has a low yield of 12.2%. It cannot be understated that this paper shows that there is a significant enrichment for dnSVs for probands without a SNV diagnosis. Furthermore, this paper shows that 34% of the pathogenic SVs cannot be detected by array-based/ES. This paper also provides the community with a new pipeline for detecting and filtering dnSVs. Moreover, 22% of the *de novo* copy-number variants once thought to be simple are complex instead. This is consistent with the observations in Schuy et al. 2022 Trends Genet Nov;38(11):1134-1146 who analyzed specific genomic syndromes, some of which show an even higher number of complex rearrangements in specific disease loci.

This is an important manuscript from one of the largest and well-characterized rare disease projects currently ongoing. Said that, the manuscript seems to have been written in a hurry without

providing key information regarding methods and results to allow other groups to repeat similar analysis. We are left wondering whether the complex SVs were resolved or not, what are the limitations of the study, how many estimated complex SVs may explain the phenotype and what were the criteria for that. Moreover, the analysis was performed using concepts developed in the cancer field which may or may not represent findings in rare diseases. In all, the manuscript to be able to communicate with the rare disease field and therefore it will benefit from a revision to clarify concepts, analysis, etc. If that is done it will provide exciting findings for the community and will help to move the field forward.

The authors are using a terminology that is distinct different from the one used in the rare disease field, including the definition of complex genomic rearrangements (for example of CGR definition see Schuy et al. Trends Genet Nov;38(11):1134-1146 and Liu et al. Curr Opin Genet Dev 2012 Jun;22(3):211-20) which may be acknowledge. The comparison between the terms would be helpful and help understanding of the important limitations of this study. Moreover, the analysis here focuses on structural variants only and does not access copy-number variation which is a key limitation to access pathogenicity in addition to limit validating the structure.

Thanks for this insightful comment and the references. Based on the reviewer’s suggestions we have decided to primarily use a nomenclature for types of complex structural variants that are used in rare disease fields (please see below; **Rebuttal Table 1**).

Original manuscript	Revised manuscript
Dup-Trp-Dup	DUP-TRP/INV-DUP
Dup-invDup	DUP-NML-DUP
Local-3-jumps	DUP-NML-DUP-NML-DUP

Rebuttal Table 1 | revised terminology to match the one used in the rare disease field

Another new addition in our manuscript is called dnCNVs using a read-depth-based algorithm (i.e.,CANVAS), resulting in an additional 286 high-confidence dnCNVs with >10kb (138 DELs and 148 DUPs). We assessed the pathogenicity of these CNVs concerning the proband’s phenotype and identified 19 pathogenic dnCNVs (see criteria below; the new additions are in Italics).

Line:430

“The identified SVs disrupting exons were reviewed for potential clinical relevance by NHS clinical scientists and/or Genomics England. We considered SVs as being potential (likely) pathogenic SVs if at least one of the following criteria were fulfilled: i) the variant had been clinically assessed as “likely pathogenic” or “pathogenic” by an NHS genomic

laboratory hub. *In detail, the variant had been assessed by an NHS clinical laboratory according to the best practice guidelines recommended by the Association for Clinical Genomic Science (ACGS) as being likely pathogenic or pathogenic and related to the primary phenotype for which the participant was recruited to the 100,000 Genomes Project. ii) the variant had been reviewed on a research basis and considered to be a strong candidate diagnostic variant. In the research review, variants resulting in loss of function for genes in which haploinsufficiency is a known mechanism of disease or variants resulting in loss of a critical domain impacting genes associated with a phenotype relevant to the primary clinical indication were considered as candidate diagnostic variants. For variants impacting genes associated with recessive phenotypes, additional variants were not considered.*”

To further assess pathogenicity of dnSVs, we validated an additional 10 pathogenic dnSV candidates using RNA-seq data. For this, the effect of dnSVs on mRNA was validated by confirming abnormal RNA-seq reads supporting dnSVs, underexpression, or aberrant splicing patterns. In particular, the degree of significance of underexpression and aberrant splicing patterns was evaluated using about 5,500 background samples.

Supplementary Figure 3 | Validation of pathogenic dnSVs using RNA-seq. (a) Evidence of WGS (top) and RNA-seq reads (bottom) supporting dnSVs. (b) Validation of underexpression of genes hit by dnSVs. The average expression of 41 genes within the deletion (i.e., 1p36) was computed (bottom), and then the degree of significance of underexpression was estimated using 5,546 background samples in Genomics England. (c) Validation of abnormal splicing patterns caused by dnSVs. The ratio was computed using abnormally- (nominator) and normally-spliced reads (denominator) and then the degree of significance of the ratio was calculated using 5,546 background samples in Genomics England.

General comment

R1.1 There is no description of the cohort. Please clarify how many probands were included, how many trios, siblings, how many males, females, etc. Methodology starts with the SV calling but there is no information about the short-reads, coverage, library. In the figures we learnt that long-reads were used but no descriptive information about the methods is provided. How many samples had long-reads in addition to short-reads?

We apologise for this oversight. We have now added this information as a **Supplementary Table 1** (please see below) describing the cohort information.

		Number		
		Male	Female	Total
Trios	Proband	7,755	5,943	13,698
	Unaffected sibling	104	107	211
Families		NA	NA	12,574
Participants (total, across all families)		20,438	18,628	39,066

Supplementary Table 1 | Cohort description

We also added a detailed description of short- and long-read data in the **Method** section.

Line:348

“Genomic blood DNA libraries were prepared using the Illumina TruSeq PCR-free protocol and samples were whole-genome sequenced on the Illumina HiSeq X platform (2 × 150-bp paired-end reads).”

Line:372

“Long-read sequencing data (*23 resequenced samples with PacBio technology*), RNA-seq (whole blood RNA sequencing from 5,546 samples) and diagnostic SNV/Indels were obtained from Genomic England (Turro et al., 2020)”

R1.2 Please clarify further how the group evaluated the clinical relevance of dnSVs. The description provided is extremely vague and nonspecific.

We added a detailed description of how we evaluated the clinical relevance of dnSVs in the **Method** section (please see below; the new additions are in italics).

Line:430

“The identified SVs disrupting exons were reviewed for potential clinical relevance by NHS clinical scientists and/or Genomics England. We considered SVs as being potential (likely) pathogenic SVs if at least one of the following criteria were fulfilled: i) the variant had been clinically assessed as likely pathogenic or pathogenic by an NHS genomic laboratory hub. *In detail, the variant had been assessed by an NHS clinical laboratory according to the best practice guidelines recommended by the Association for Clinical Genomic Science (ACGS) as being likely pathogenic or pathogenic and related to the primary phenotype for which the participant was recruited to the 100,000 Genomes Project.* ii) the variant had been reviewed on a research basis and considered to be a strong candidate diagnostic variant. *In the research review, variants resulting in loss of function for genes in which haploinsufficiency is a known mechanism of disease or variants resulting in loss of a critical domain impacting genes associated with a phenotype relevant to the primary clinical indication were considered as candidate diagnostic variants.*”

R1.3 Please consider adding a workflow for the pipeline and filtering process used to identify the dnSVs.

Thanks for this suggestion, we have now generated a workflow of our pipeline for the identification of dnSVs (please see below; **Supplementary Figure 13**). Plus, we added the number of the passed SVs after each filtering step in the workflow.

Supplementary Figure 13 | Flowchart for dnSVs pipeline.

R1.4 CNVs constitute a very important part of the majority of CGRs in rare diseases. In fact, most of the clinical phenotype will arise from haploinsufficient or triplosensitive loci. It seems that only SVs disrupting exons were reviewed for potential clinical relevance which will miss a lot of potentially pathogenic SVs.

We appreciate the reviewer's comment regarding the role of CNVs in CGRs and their contribution to rare disease phenotypes. We acknowledge that many clinical phenotypes arise from haploinsufficient or triplosensitivity loci, and we would like to clarify our approach to evaluate pathogenic SVs in our study. In the original manuscript, we assessed SVs that disrupt exons, which also included whole or partial gene deletions. Our primary criterion for classifying an SV as potentially pathogenic was whether it affected a known dominant disease-associated gene, particularly those classified as haploinsufficient. Therefore, haploinsufficient genes were systematically evaluated for their potential clinical relevance, as they represent a form of genetic dominance where gene dosage is critical. However, we acknowledge that our approach to assessing

triplosensitive genes was more conservative than our comprehensive evaluation of loss-of-function dnSVs in dominant genes. Specifically, we limited our assessment of triplosensitive genes to those (n=7) with strong evidence for dosage pathogenicity, as curated in:

- ClinGen dosage sensitivity maps; genes classified as “Sufficient” or having some “evidence for dosage pathogenicity.
- DDG2P; genes classified as “Definitive”

In this cohort, we did not identify de novo duplications affecting genes classified as triplosensitive under these stringent criteria. However, we recognise that further exploration of triplosensitivity in CGRs is an important area for future follow ups. To address the reviewer’s concern, we have now explicitly stated in the Manuscript that our analysis of pathogenic SVs includes whole or partial gene deletions affecting haploinsufficient genes. Additionally, we have clarified in the Discussion section that identification of triplosensitive genes contributing to patient phenotypes remains an area for future investigation.

Line:298

“Finally, *the systematic identification of gene duplication leading to triplosensitivity (Collins et al., 2022)* and inherited pathogenic SVs will be needed to facilitate a more comprehensive diagnosis.”

R1.5 Please add more information about the pathogenic SVs not detected by array or exome. A table with the type of rearrangement, sizes, genes affected would be helpful.

Thanks for this suggestion. We previously defined CNVs **affecting < 2 exons** as not likely to be detected by array-based or exome-seq methods. However, it’s generally recognised that reliable detection of CNVs from WES can be accomplished when CNVs span at least 3 exons (Krumm et al., 2012; Pfundt et al., 2017). Therefore, we defined CNVs **affecting < 3 exons** as not likely to be detected by array-based or exome-seq methods in the revised manuscript. As a result, 66 of the 145 pathogenic SVs (46%) are not likely to be detected by array-based or exome-seq methods. These undetectable SVs include balanced rearrangements (i.e., reciprocal inversion or translocation; n=19), which are not accompanied by copy number changes, and deletions affecting < 3 exons (n=47; median = 4.5kb; see the method description below). We have flagged which pathogenic dnSVs are likely to be detectable/undetectable by array/exome-seq methods in **Supplementary Table 2** (13th column).

Line:443

We classified pathogenic SVs annotated with “reciprocal inversion” or “reciprocal translocation” as not likely to be detected by array-based or exome-seq methods. In addition, the SVs involving deletions spanning < 3 exons (Krumm et al., 2012) of GENCODE canonical transcript were classified into this undetectable category.

R1.6 Why were the coordinates converted from hg38 rather than the BAM remapped to hg38? I can see the advantage with the remapping, but it is not clear what is the advantage of the liftover. In fact, I am wondering whether that could add inconsistencies to the mapping and analysis, especially for CGRs.

We appreciate the reviewer’s comment and fully agree with their technical concerns. To clarify, we only performed coordinate conversion from hg19 to hg38 when analysing the distribution of dnSVs concerning telomere ends (**Figure 4b**). The primary reason for this approach was to ensure consistent alignment with telomere annotations, which are more accurately defined in the hg38 reference genome. However, we resolved the genomic configuration of complex genomic rearrangements (CGRs) directly in hg19-mapped samples, without liftover, as liftover could introduce positional inconsistencies that might affect breakpoint interpretation. To maintain consistency and clarity we have provided dnSVs with their original hg19 coordinates in **Supplementary Table 2**.

R1.7 Please clarify what it means to have the SVs “manually” validated. I have not seen alignment of the breakpoint junctions which is usually the analysis that can be performed using split reads from IGV to confirm: 1. the structure of the CGR; 2. Whether the CGR was fully resolved. If junctions are still open ended, that means the structure was not resolved. Relevant for that is the fact that we often identify real pathogenic SVs that have no split reads although they have discordant read pairs. This happens when the breakpoint junction maps to repeats (SINEs, LINEs, etc). In those cases, Sanger or Target ONT (MinION works well), need to be used to resolve the structure. There was no comment about those type of SVs, were they filtered out? In addition, CGRs often have insertions that are larger than the short-reads and can not be resolved without long-reads or long-read PCR. Please clarify how the analysis was performed and data filtered regarding those issues.

We totally agree with this insightful comment and apologise for the lack of clarity regarding how we performed manual validation. As the reviewer correctly pointed out, breakpoint resolution and validation are critical for confirming CGRs. From a technical perspective, we considered three types of detectable SVs based on their sequencing evidence. The first two were included in the original manuscript, and the third type has now been explicitly addressed in the revised version. For all three categories, manual validation involved confirming the absence of abnormal reads or read depth changes in the proband’s parents.

1. **SVs supported by discordant and split reads:** these SVs were manually validated by inspecting the presence of both discordant read pairs and split reads supporting the SV in the proband.
2. **SVs supported only by discordant reads:** these SVs were tagged as “IMPRECISE” by Manta, indicating imprecise breakpoints due to the absence of split reads (as the reviewer pointed out). To improve confidence in these calls, we required additional read-depth support, flagging only those variants that were also “ColocalizedCanvas” (i.e., supported by read-depth analysis). We manually validated the presence of discordant reads and read depth changes.
3. **SVs (i.e., CNVs) supported only by read depth:** these were visually validated by inspecting read-depth changes to confirm their presence in the proband.

We added more description regarding this process in the method section to better describe our manual inspection strategy.

Line:361

“We rescued SVs with imprecise breakpoints if they were supported by CANVAS. In detail, SVs tagged as “IMPRECISE” (i.e., imprecise breakpoints) were called if they were also flagged as “ColocalizedCanvas”.

Line:366

“All SVs were manually validated to identify high-confident *de novo* SVs using IGV browser (Robinson et al., 2011). Specifically, we visually validated the presence of abnormal reads (i.e., discordant/split reads) in probands and the absence of such reads in their parents.”

We also acknowledge the limitation of detecting complex genomic rearrangements using short-read sequencing in the discussion section.

Line:286

“We note several limitations, which open potential venues for future investigations. Complex structural variants are still underestimated in our cohort due to the inherent limitation of short-read-based SV discovery. For example, SVs in repeat-rich regions (e.g., segmental duplications

or retrotransposons) remain challenging to identify. Although we identified some of such SVs using read-depth-based algorithms (e.g., CANVAS), resolving the genomic configuration of complex SVs using read-depth-specific calls that don't provide read orientation along with read-pattern-based calls is challenging. Furthermore, short-read sequencing is known to fail to capture large SVs, especially large insertions.”

R1.8 In the “SV calling” session is written that “SVs found in 3+ samples were removed because such SVs were likely to be germline SVs”. This may be an incorrect statement since the analysis is looking for germline SVs.

Apologies for this incorrect description. We corrected this as follows.

Line:358

“SVs found in 3+ samples were removed because such SVs were likely *alignment artifacts*.”

R1.9 SVs with VAF < 0.1 were excluded as potential mosaicism. However, it may also be excluding real ones in low coverage regions. Have you checked for that?

Thanks for raising this point regarding the potential exclusion of true *de novo* SVs in low-coverage regions. To ensure that we did not inadvertently exclude true dnSVs, we carefully examined the ten SVs (five deletions, four tandem duplications and one inversion) that were removed due to their VAF being less than < 0.1. We computed VAF using the PR tag from MANTA, which represents the number of spanning read pairs that support REF and ALT (Q30). The median read depth (REF + ALT) of these ten excluded SVs was 71.5 (range:61 -132), indicating that they were located in well-covered regions (see table below).

Chr	Start	End	Type	VAF	Coverage
5	175809650	175837315	DUP	0.08	72
2	211941492	211943205	DEL	0.09	61
5	37062749	37430927	DUP	0.09	106
6	65712959	65739706	INV	0.08	64
2	141469311	141780465	DEL	0.09	111

22	39997497	40006388	DEL	0.08	132
7	9105470	9108674	DUP	0.08	88
7	25992106	26104091	DUP	0.08	62
19	28642835	28644590	DEL	0.08	70
2	50766228	50857001	DEL	0.07	71

Rebuttal Table 2 | Excluded potential mosaic SVs

Furthermore, none of these excluded SVs were associated with probands' phenotype-relevant genes supporting the likelihood that they were low confidence or mosaic events rather than true germline dnSVs. Given these observations, we believe that retaining the VAF threshold of 0.1 is appropriate to exclude potential false positives and to ensure high-confidence dnSV detection. We have now explicitly stated the number and characteristics of the filtered SVs in the Methods section (please see below).

Line:363

“We excluded SVs with VAF < 0.1 (n=10) to remove potential mosaic SVs.”

R1.10 Please define clustered breakpoints and explain the benefit of using ClusterSV in CGRs for rare diseases and how many CGRs were detected by using this program. This is relevant because most of the rare diseases CGRs do not cluster (except for a few rare cases of chromothripsis and show breakpoints that map far away from the CNV/SV. They are often copied from regions 20 kb or less away, sometimes from distinct chromosomes. Please see discussion from Carvalho et al. 2013. Nat Genet. 2013 Nov;45(11):1319-26).

We agree with the reviewer's point that most rare disease-associated CGRs do not always form tightly clustered breakpoints, except in rare cases such chromothripsis (Carvalho et al., 2013). ClusterSV is a sophisticated algorithm for grouping rearrangements into rearrangement clusters and footprints. The algorithm is detailed in supplementary note (pages 27-31) in PMID: 3202501. Supplementary note link :

https://static-content.springer.com/esm/art%3A10.1038%2Fs41586-019-1913-9/MediaObjects/41586_2019_1913_MOESM1_ESM.pdf

The key advantage of ClusterSV in studying CGRs for rare diseases is that it identifies clusters of SV breakpoints without requiring a predefined distance threshold, allowing for a data-driven

detection of complex genomic rearrangements. It can capture CGRs with dispersed breakpoints, even those spanning hundreds of kilobases, as long as they are statistically enriched compared to expectation. Lastly, it helps to classify CGRs beyond simple CNV-based definitions, allowing diverse SV types that may otherwise be missed. In our study ClusterSV identified 158 CGRs, suggesting that this approach is comprehensive for detecting both clustered and dispersed CGRs. It is worth noting, the furthest distance between clustered breakpoints identified using ClusterSV in our study was observed in a DUP-TRP/INV-DUP event (**Rebuttal Figure 1** below), where the breakpoints spanned ~920kb (i.e., between SV1 and SV2; **Rebuttal Figure 1**). This suggests that ClusterSV can successfully detect CGRs with a broad range of breakpoint distances, including those where rearrangement patterns involve multiple SVs over large genomic intervals. Therefore, while not all rare disease CGRs exhibit tight breakpoints clustering, ClusterSV remains a useful tool for identifying complex structural variants even in cases where breakpoints are moderately dispersed.

Rebuttal Figure 1 | Example of DUP-TRP/INV-DUP identified by ClusterSV. The distance between the two different inversions (SV1 and SV2) was ~920kb.

We added a description of the advantage of ClusterSV in the method section.

Line:386

“We used ClusterSV(Li et al., 2020) (<https://github.com/cancerit/ClusterSV>) to group rearrangements (i.e., breakpoints), into rearrangement clusters. *The key advantage of ClusterSV is*

to identify clusters of dispersed breakpoints (Carvalho et al., 2013) without requiring a predefined distance threshold, allowing for a data-driven detection of complex genomic rearrangements.”

R1.11 What are dispersed duplications? In rare diseases we often see duplications separated by copy-number neutral regions (by CNV analysis) which are often overlooked as CGRs because they can be far apart. However, analysis of junctions shows they are connected, and that are part of the same SV event. Are those the same type? Please see reference here for more details on this type of structure: Gu et al. 2015 Hum Mol Genet. 2015 Jul 15;24(14):4061-77. doi: 10.1093/hmg/ddv146 and Liu et al. 2017 Cell. 2017 Feb 23;168(5):830-842.e7. doi: 10.1016/j.cell.2017.01.037.

Dispersed duplications (i.e., interspersed duplications) occur when a duplicated sequence is inserted at a different genomic location rather than remaining adjacent to the original copy. We have now included a schematic representation of this dispersed duplication in **Supplementary Figure 8** (please see below).

Supplementary Figure 8 | Schematic of representative complex dnSVs classes. The pink segments represent duplicated regions.

R1.12 “Local 3-jumps” is stated as not yet observed in rare diseases, I do not think this this is correct as from the plots they look like separated CNVs connected via junctions which correspond to the definition of DUP-NML-DUP or DUP-NML-DUP-NML-DUP used in several papers. It seems that the nomenclature for cancer and rare disease CGRs are different which makes it difficult to compare structure, please comment. Beck et al. Cell. 2019 Mar 7;176(6):1310-1324.e10. doi: 10.1016/j.cell.2019.01.045, Bahrambeigi et al. Genome Med. 2019 Dec 9;11(1):80. doi: 10.1186/s13073-019-0676-0, Carvalho and Lupski, Nat Rev Genet. 2016 Apr;17(4):224-38

We apologise for the nomenclature confusion and appreciate the reviewer’s clarification. As the reviewer correctly pointed out, the structural pattern referred to as “Local 3-jumps” in cancer genomics corresponds to the DUP-NML-DUP-NML-DUP structures described in rare disease

structural variant literature (Bahrambeigi et al., 2019; Beck et al., 2019; Carvalho and Lupski, 2016)

To ensure consistency, we have updated the manuscript to adopt the DUP-NML-DUP-NML-DUP terminology and have explicitly referenced the rare disease literature where similar complex genomic rearrangements (CGRs) have been observed.

Line:180

“Interestingly, beyond local-2 jumps (i.e., clusters of two rearrangements) as described above, we also found three instances of ‘DUP-NML-DUP-NML-DUP’ (Bahrambeigi et al., 2019; Beck et al., 2019; Carvalho and Lupski, 2016) (i.e., known as local-3-jumps in the cancer field (Li et al., 2020)) involving three local rearrangements (**Figure S9A**).

We also found that “DUP-NML-DUP” was referred to “Dup-invDup” in our original manuscript (see **Reviewer Figure 2** below). We renamed “Dup-invDup” as “DUP-NML-DUP” in the revised version.

Line:175

“The other remaining classes, comprising duplication and inversion, are ‘DUP-NML-DUP’ (Gu et al. 2015; Schuy et al. 2022) (i.e., paired duplication inversion/Dup-invDup; n=7; 4.7%; **Figure S7**) and ‘DUP-TRP/INV-DUP’ (i.e., Dup-Trp-Dup; n=14; 9.5%; **Figure S7**), exhibit structures involving two duplications linked by inverted rearrangements and duplication–inverted-triplication–duplication, respectively.”

Reviewer Figure 2 | Schematic representation of DUP-NML-DUP (**Figure 1C** from (Schuy et al., 2022)) and Dup-invDup (**Figure 4A** from (Li et al., 2020)).

R1.13 Line 445: I suggest the author organize and document the code. Please explain in the README what this pipeline does, what are the expected inputs/outputs. Please also make this command-line executable so that users can use this tool easily.

We restructured the code repository and added more comments in the codes (e.g., expected inputs/outputs).

Results

R1.14 It will be helpful to see the size distribution of all SVs identified in this cohort. Is there a bias towards identifying smaller SVs? It looks like deletions are mostly small.

The median detected deletion size in our cohort is 3.7 kb (range:52bp - 61mb; see **Reviewer Figure 3**, below). We compared the deletion size between ours and the previous study and found no significant difference (p-value = 0.22; two-sample Kolmogorov-Smirnov test).

Reviewer Figure 3 | Comparison of the distribution of deletion length between our cohort and the previous study (PMID : 33675682)

Below we plotted the size distribution of 7 SV types with at least 10 events (see **Reviewer Figure 4**).

Reviewer Figure 4 | Size distribution of 7 SV types with at least 10 events (SV types involving a translocation were not included).

R1.15 Please consider adding the distribution of calls per proband across the cohort (maybe a boxplot)

We plotted the distribution of calls using probands with at least one dnSV (see figure below). We have decided to use a histogram rather than a boxplot because the former plot is more effective for displaying the overall distribution of the calls. For the reviewer's information, the addition of CNV calls by CANVAS (n=286) to our original calls didn't change individuals with multiple dnCNVs phenotype.

Supplementary Figure 1B | Histogram of the number of individuals (y axis) by the number of *de novo* SVs (x axis) using probands with at least one dnSV (n=1,696).

Line:85

Using 1,870 high-confidence dnSVs from 1,696 probands (91% of probands had a single SV; **Figure S1B**), we estimated an overall mutation rate of 0.13 events per genome per generation, in line with previous reports^{24–26} (**Figure S4A**).

R1.18 Line 326: very stringent filtering, going to miss recurrent CNVs, even if it is *de novo*. It is not totally clear why read-depth was not used to check for CNVs in this cohort. Agree that small SVs (smaller than 10 kb) are hard to identify and would call a lot of false-positive CNVs, but they can still be confirmed once identified by Manta. Larger SVs should certainly be checked.

As the reviewer suggested, to complement our original SV calls, we expanded our analysis to include CNVs **≥10kb** using the read-depth-based algorithm CANVAS, as the reviewer suggested. This additional analysis identified 286 CNVs (138 DELs and 148 DUPs) across 258

samples, which were then manually validated. We also assessed the pathogenicity of genes impacted by the CNVs concerning the probands' phenotype, resulting in 19 pathogenic dnCNVs (**Supplementary Table 4**). These read-depth-specific calls were only used to assess their pathogenicity because clustering and classification (i.e., resolving genomic configuration of complex SVs) of these calls that didn't provide read-orientation are challenging. As for the smaller SVs (smaller than 10kb), we already identified them using Manta in the original manuscript. ($\geq 50\text{bp}$).

R1.19 Please provide a table with the SVs that were classified as pathogenic/potentially pathogenic (criteria) or VUS, type, size, coordinate, and detailed information about genes affected by the SVs, disease, phenotype of patient. I would suggest a table with the pathogenic CGRs as part of the main text.

Thank you for this suggestion. In the revised manuscript, we have opted to report only pathogenic or likely pathogenic dnSVs. As a result, we have removed six dnSVs previously classified as VUS, resulting in 145 pathogenic dnSVs. To provide a clear and comprehensive overview, we have included all identified dnSVs, including pathogenic ones, with SV type, coordinate, affected genes, and probands' disease categories in **Supplementary Table 2**.

R1.20 Was only 2.4% (44 out of 1872) of the SV calls validated by an orthogonal technology? If a SV affects a dosage sensitive gene potentially contributing to the phenotype, was it validated by array, exome, or ddPCR? If disrupting a dosage sensitive gene, was it validated by Sanger or RNA expression? Please clarify.

Although we validated 2.4% of the calls using an orthogonal technology (i.e., array or long read), as we described earlier, we performed a systematic visual validation of each SV. While this subset represents a small fraction of the total SVs, these validations were aimed at ensuring overall call accuracy across representative variant types. Every identified SV underwent visual inspection in IGV to make sure of high confidence calls. Furthermore, 145 pathogenic dnSVs were independently reviewed by an external team at Genomic England.

In the revised manuscript, we performed an additional RNA-Seq validation of 10 pathogenic dnSVs (10/145; 7%). Specifically, we aimed to confirm the functional impact on gene expression by assessing: abnormal RNA-seq reads (n=4; **Supplementary Figure 3a**) supporting SVs, significant underexpression of affected genes (n=2; **Supplementary Figure 3b**), or aberrant splicing patterns resulting from dnSV induced disruption (n=4; **Supplementary Figure 3c**).

Supplementary Figure 3 | Validation of pathogenic dnSVs using RNA-seq. (a) Evidence of WGS (top) and RNA-seq reads (bottom) supporting dnSVs. (b) Validation of underexpression of genes hit by dnSVs. The average expression of 41 genes within the deletion (i.e., 1p36) was computed (bottom), and then the degree of significance of underexpression was estimated using 5,546 background samples in Genomics England. (c) Validation of abnormal splicing patterns caused by dnSVs. The ratio was computed using abnormally- (nominator) and normally-spliced reads (denominator) and then the degree of significance of the ratio was calculated using 5,546 background samples in Genomics England.

R1.21 In rare diseases, segmental duplications are often involved in the formation of de novo SVs, CNVs, and CGRs. But they will be missed by Manta and any other variant caller that do not use read-depth. Alus and LINES are missed sometimes often too, especially if Alu-Alu or LINE-LINE forming a fusion repeat. Those are all limitations of the approach used here and should be acknowledged. CGRs may be still underestimated in this cohort.

Thanks for this insightful comment. We fully acknowledge the limitations of short read sequencing in detecting CGRs, particularly in repeat rich regions. We complemented our Manta calls with CANVAS CNV detection, which captures CNVs that may be missed by read-pair/split-read approaches. However, CANVAS lacks breakpoint resolution and read orientation, making it difficult to fully resolve CGRs. As a result some CGRs may still be underestimated, particularly those involving complex breakpoint structures or large insertions, which are known limitations of short-read sequencing. To improve clarity about these limitations, we have updated the Discussion section to explicitly state these limitations (please see below).

Line:286

“We note several limitations, which open potential venues for future investigations. While integrating read-depth (e.g., CANVAS) and read-pattern based (e.g., Manta) approaches improves SV detection, limitations remain in resolving CGRs particularly those mediated by segmental duplications or repeat-rich regions (e.g., segmental duplications or retrotransposons). Additionally, short-read sequencing is inherently limited in detecting large insertions and complex SVs requiring large-range phasing”

R1.22 Triplications are very important CGRs in rare diseases, so it is really exciting that the group found 41. They do deserve a bit more description and discussion. Are the 41 triplications of the same type? If they are all DUP-TRP/INV-DUP, what are the sizes of the segments? Please provide results to exemplify. Importantly, depending on the structure and gene content it will affect

expression in distinct ways. It can cause a more severe phenotype due to higher expression, disrupt a gene or lead to fusion genes (detailed discussions here Grochowski et al. 2024 Cell Genom. 2024 Jul 10;4(7):100590. doi: 10.1016/j.xgen.2024.100590). They can also cause other alterations depending on the mechanism such as imprinting disease (Carvalho et al. Genome Med. 2019 Apr 23;11(1):25).

We would like to apologise for potentially misusing the term “triplications”. As the issue was also raised by Reviewer 2 (which has responded in length in **R2.22**), in the original manuscript, we used the “triplication” term to describe genomic loci where the total copy number is 3, consisting of one normal copy allele (copy number = 1) and duplicated allele (copy number = 2). To ensure accuracy, we now restricted the term “triplication” to specifically describe DUP-TRP/INV-DUP events (previously referred to as “Dup-Trp-Dup” in the original manuscript). We identified 14 DUP-TRP/INV-DUP events in our cohort. The median size is 334 kb (range:61kb-930kb; **Supplementary Figure 7d**).

Supplementary Figure 7d | Size distribution of DUP-TRP/INV-DUP identified in this cohort (left) and an example of DUP-TRP/INV-DUP with a length of 303kb (right).

In the revised manuscript, we have expanded our discussion on the potential pathogenic mechanisms of DUP-TRP/INV-DUP.

Line:184

“While a majority of the pathogenetic effects of all these complex SV types involving duplication are related to the overexpression of triplosensitive genes (Bahrambeigi et al., 2019; Grochowski et al., 2024) (i.e., gain-of-function), these variant types have been reported to cause disease by

loss-of-function mechanisms such as gene disruption (Ishmukhametova et al., 2013), gene fusion at breakpoints (Zuccherato et al., 2016), and segmental uniparental disomy (Carvalho et al., 2019).”

R1.23 Regarding the mechanism of triplication formation, it is not clear why it was assumed to be a meiotic event upfront. It seems that the analysis is basically phasing of the SNVs, is that correct? If yes, the only information obtained is the origin of the rearranged variant. There are no results provided (for example, how many SNVs per duplication/triplication are informative?) Moreover, triplications were shown to occur in mitosis (paternal germline Carvalho et al. 2011 Nat Genet. 2011 Oct 2;43(11):1074-81. doi: 10.1038/ng.944) or during development with mixed parental origin for TRP and DUP (Carvalho et al. 2015 and references within Am J Hum Genet. 2015 Apr 2;96(4):555-64. doi: 10.1016/j.ajhg.2015.01.021; Liu et al. 2017 Cell. 2017 Feb 23;168(5):830-842.e7. doi: 10.1016/j.cell.2017.01.037.)

The reviewer is correct. We used a set of phased SNVs to infer the timing of maternally derived duplication formation. We provided the details of the duplication (e.g., size and the number of available SNPs) used in the timing analysis in **Supplementary Table 3**.

We only attempted to infer the timing of duplication formation in maternal origin because paternal duplication can occur in premeiotic state during male gametogenesis throughout life, as the reviewer correctly pointed out (Carvalho et al., 2011). We explained the reason why we focused only on maternal duplication in the revised manuscript.

Line:423

“The timing of paternally-derived duplications was not inferred due to the fact that duplications can also occur in a premeiotic state during male gametogenesis throughout life (Carvalho et al., 2011).”

We also excluded duplications for which the parent of origin was inferred as maternal and paternal origin (i.e., those occurring during development with mixed parental origin for TRP and DUP). We stated this filtering criterion in the method section as follows.

Line:417

*“To classify the timing of maternally-derived duplication into meiosis I and II, we first identified duplication (including those in complex SVs) from maternal origin (step 1) and further classified them into meiosis I and II (step 2) using a set of informative genotypes (**Figure S6**). For binary classification at each step, the ratio of the number of SNPs supporting one class to another class*

was calculated, and a class for which the ratio was greater than 0.9 was chosen, at least three SNPs were required for either class at each step. These filtering criteria could time large duplications with a handful of erroneously called SNPs and remove ambiguous duplications such as those originating from both parents during early development (Carvalho et al., 2015; Liu et al., 2017) (e.g., potentially due to mitotic crossing-over)."

R1.24 The finding of multiple dnSVs is very interesting. Are these samples considered to be outliers? Would that finding be validated or could be a technical artifact? If confirmed, please consider showing those results as a table with type, sizes and origin of the ancestral chromosome if available. Are they independent events (i.e., the junctions are not connected forming single SV originated in the same event)? For comparison and mechanistic discussions, multiple CNVs are extremely rare but they offer a glimpse of potential prone to error DNA repair mechanism leading also to higher frequency of SNVs: Liu et al. Cell. 2017 Feb 23;168(5):830-842.e7. doi: 10.1016/j.cell.2017.01.037; Du et al. Genome Med. 2022 Oct 27;14(1):122; Beck et al. Cell. 2019 Mar 7;176(6):1310-1324.e10.)

Thanks for listing these papers for us to learn more about the multiple dnCNVs phenomenon. Because individuals with ≥ 4 dnCNVs were defined as multiple dnCNVs phenotype in these reference papers, we increased the minimum number of SV thresholds from 3 to 4 to define individuals with multiple dnCNVs, resulting in four individuals (see table below; the details can be found in Supplementary Table 2). These dnSVs were visually validated and seemed to occur independently (i.e., not clustered). Unlike the known multiple dnCNVs phenomenon that shows a predominance of copy number gain, 88% of the identified dnSVs are a deletion (median = 1.5kb), suggesting that further investigation is needed to characterize the individuals with multiple CNVs in our cohort. We discussed our findings with the reference papers in the revised manuscript as follows.

Line:94

“We identified four individuals with a considerably higher number of dnSVs ($n \geq 4$). These individuals, recruited under different rare disease categories, are not among the previously reported germline SNV hypermutators in this (Kaplanis et al., 2022) cohort and have no known history of parental exposure to chemotherapy. Unlike the known multiple dnCNVs phenomenon that shows a predominance of copy number gain (Du et al., 2022; Liu et al., 2017), 88% of the identified dnSVs in these individuals were a deletion (median = 1.5kb), suggesting that further investigation is needed to characterize the multiple dnSVs in these individuals.

chr	start	end	type	length	De-identified-sample-ID
1	120705649	120947405	MantaDUP	241756	1641
10	39256689	39268392	MantaDEL	11703	1641
6	68531577	68533151	MantaDEL	1574	1641
6	75410626	75432929	MantaDUP	22303	1641
9	87022397	87027846	MantaDEL	5449	1641
12	113832285	113833849	MantaDEL	1564	1641
16	79803574	79804956	MantaDEL	1382	1641
X	44718884	44719306	MantaDEL	422	1641
3	75374117	75593452	MantaDEL	219335	1641

8	11541670	11544145	MantaDEL	2475	178
8	46641941	46642024	MantaDEL	83	178
8	124204160	124208559	MantaDEL	4399	178
13	97952373	97955865	MantaDEL	3492	178
17	16299046	16308188	MantaDEL	9142	178
17	54241549	54249429	MantaDEL	7880	178
19	51962930	51969570	MantaDEL	6640	178

1	235287980	235295151	MantaDEL	7171	1489
2	97096720	97098182	MantaDEL	1462	1489
2	122936243	122936297	MantaDEL	54	1489
15	64141519	64141703	MantaDEL	184	1489

2	167773489	167773835	MantaDEL	346	1629
18	62579407	62579993	MantaDEL	586	1629
18	69193958	69194522	MantaDEL	564	1629

12	47990260	50068864	MantaBND	NA	1629
----	----------	----------	----------	----	------

Rebuttal Table 3 | Four individuals with multiple dnCNVs. Only two SVs could be phased to determine the parent of origin (**Supplementary Table 2**).

Minor comments

R1.25 Line 261: should probably also mention that chrY is excluded in this study

Thanks for pointing this out. We mentioned this as follows.

Line:354

“We first removed SVs on the Y chromosome and further removed SVs having evidence of clipped reads (i.e., split reads) at breakpoints in either parent.”

R1.26 Line 330: what is MGE10kb?

“MGE10kb” is a filtering flag defined by Manta. It represents “Manta calls with length < 10 kb”. We didn’t filter out SVs with this flag, because we wanted to analyze SVs with a length >= 50bp. We added a detailed description of “MGE10kb” as follows.

Line:358

“We selected SVs flagged as “PASS” or “MGE10kb” (i.e., Manta calls with length < 10 kb) and further narrowed down SVs with the Manta score > 30 and supporting discordant reads > 10.”

R1.27 Line 368: Please check the documentation for the program Unfazed, it should be extended read-backed phasing, not read-based phasing.

The reviewer is correct. We revised our manuscript as follows.

Line:411

*“We used unfazed, which employs both *extended read-backed- and SNV allele-balance-phasing*, to identify the parent of origin for dnSVs.”*

R1.28 Figure1: b) cannot tell how each pair is significantly different. The medians of the boxplots are visually at the same level... c) I think the x-axis is wrong. How can frequency be 1000? Log-scale?

b) Although the medians in the boxplots look similar, the third quantile values (i.e., the upper lines of the boxplots), lower-, or upper-whisker values are clearly higher in probands with dnSVs than those without dnSVs, that makes statistically significant differences between the groups.

c) The scale of the x-axis (log10 transformation) is correct. For example, deletion accounts for 73.6% of total dnSVs (1377/1872).

R1.29 Figure3: d) typo “intronic”

Thanks for spotting this. We have now corrected it.

R1.30 FigureS4: c) this does not add up to ~1800 dnSVs, what happened to the rest?

This is because some of the identified dnSVs could not be phased due to a lack of available SNPs for phasing. Our phasing analysis found that 51% of dnSVs (962/1872) were phasable. We described this phasing rate in the method section as follows.

Line:413

“Phasable dnSVs (51%; 962/1872) were used for downstream analysis.”

R1.31 FigureS5: what is R?V? (I assume these are alleles?)

Yes. “R” and “V” represent “Reference” and “Variant” alleles, respectively. We explicitly denoted this in **Supplementary Figure 6** legend.

Supplementary Figure 6 legend

“R” and “V” represent “Reference” and “Variant” alleles, respectively.

R1.32 FigureS7: should probably caveat that the shown derivative structures are not the only possibilities. Eg dup-trp-dup can have other rearrangements. The original cancer paper also shows that.

The reviewer is correct. Because our aim in this study is not to reveal novel derivative structures, we displayed one of the possible genomic configurations as an example. We clearly denoted this in the

Supplementary Figure 8 legend.

Supplementary Figure 8 legend

“The bottom rearranged segment shows one of the possible genomic configurations resulting from each dnSVs class. Note that all possible schematic configurations for each class are not shown.”

1.33 FigureS12: can we have the number of calls per step? Want to know the effect of each filtering step

Thanks for this suggestion. We have now added the number of the passed calls after each filtering step.

Supplementary Figure 13 | Flowchart for dnSVs pipeline.

R1.34 Reviewer #1 (Remarks on code availability):

We suggest the author organize and document the code. Please explain in the README what this pipeline does, what are the expected inputs/outputs. Please also make this command-line executable so that users can use this tool easily.

Our current complex SV classification pipeline is not fully automatic. (i.e., semi-automatic with manual confirmation for some complex SV types). We aim to publish another method paper on a fully automated SV classification pipeline using a bam file as input. We've provide a naive command-line executable script to classify complex SVs that don't require manual confirmation

of copy number changes (i.e., Loss-Loss, Inv-Loss, Loss-Inv-Loss, Loss-invDup, DUP-TRP/INV-DUP, DUP-NML-DUP, and Dispersed Dup) at https://github.com/hj6-sanger/GEL_SV/.

Reviewer #2 (Remarks to the Author):

The study performed by Jung et al. investigates the features of de novo structural variants (dnSVs) in a large cohort of over 13,000 offspring with rare diseases, providing a unique and substantial contribution to the field of genomics and rare disease research.

I was particularly impressed by several aspects of the study:

1. Unprecedented Dataset:

The study utilized the dnSV dataset from over 13,000 offspring in the UK Biobank, the largest of its kind to date. This extensive dataset not only enhances the statistical power of the study but also allows for a more comprehensive exploration of the properties and prevalence of dnSVs. Such large-scale analysis is crucial for uncovering subtle yet significant patterns that smaller studies might miss.

2. Insight into Complex dnSVs:

The study offers new insights into the complexity and clinical impact of dnSVs, particularly those that are challenging to resolve using traditional techniques, e.g, array-based method or exome sequencing. The authors' approach to visualizing these complex variants is exemplary, making their findings accessible and easily interpretable.

3. Clinical Relevance:

By thoroughly examining the clinical implications of dnSVs, the study provides critical information that could influence future diagnostic strategies for rare diseases. The discussion around the paternal origin of these variants and the nuances of maternal vs. paternal contributions adds depth to our understanding of genetic inheritance and its role in disease.

While the study is robust and impactful, I have a few suggestions regarding the origin analysis and interpretation of the paternal and maternal contributions.

We appreciate this excellent summarisation of our study.

R2.1 Clarification on Paternal Origin Effects:

In line 109-110, the authors reference a previous study by Kong et al., which discusses de novo single nucleotide variants (SNVs), to support their findings on the paternal origin of dnSVs. However, since SNVs and structural variants might have different properties, I recommend that the authors clarify this distinction and provide additional context on how their findings compare to existing knowledge in the field of structural variants.

We agree with the reviewer's comment that dnSVs and dnSNVs may have distinct mutational properties. Our intention was not to imply that these two variant types share the same mutational processes but rather to highlight the similarity in their parental origin bias. To avoid any potential confusion, we revised the relevant sentences in the manuscript as follows.

Line 114 :

“Additionally, we observed 67.8% of the phased dnSVs originated from paternal germ cells (**Figure S4C**), as a proportion consistent with previous studies on structural variation (66-74.4%) (Belyeu et al., 2021; Brandler et al., 2018). This finding aligns with the well-documented paternal bias in de novo SNVs and indels, reinforcing the broader trend of increased germline mutagenesis in the male lineage (Kong et al., 2012) ”

R2.2 Discussion on Maternal vs. Paternal Origins:

The manuscript could benefit from a more detailed discussion of the differences between maternal and paternal origins of dnSVs, especially regarding triplications/de novo duplication (line 126-129). Addressing whether paternal origin de novo duplications can be inferred to meiosis I or II and discussing the ratio between maternal and paternal contributions could offer additional insights, at least for the duplication. In addition, the use of the term ‘triplication’ might need some clarification. It appears that the term is used to describe genomic loci where the total copy number is 3, with one normal copy allele (copy number = 1) and a mutated allele with a duplication (copy number = 2). However, "triplication" is also commonly used to refer to an allele with three copies, such as in a duplication–inverted-triplication–duplication structure (line 164). Clarifying this terminology in the manuscript could help prevent potential confusion for readers.

We appreciate the reviewer's suggestion to expand on the differences between maternal and paternal origins of *de novo* duplications. This was also raised by Reviewer 1 (R1.22), and we have also provided a more detailed response there. Our timing analysis was limited to maternally-derived duplication because paternal duplications can also occur pre-meiotically during spermatogenesis, which occurs continuously throughout life. As a result, paternal duplications cannot be reliably classified into meiosis I or II using SNP phasing data (please see below).

Line 423 :

“*The timing of paternally-derived duplications could not be inferred due to the fact that duplications can also occur in a premeiotic state during male gametogenesis throughout life.*”

We added a background of our timing analysis in the manuscript and provided the details of duplications used in the timing analysis in the **Supplementary Table 3**.

Line 134 :

“We inferred the timing of maternally-derived duplication formation into meiosis I and II based on the fact that heterologous allele duplications are known to occur only before the separation of homologous chromosomes during meiosis I, while homologous allele duplications are known to occur before the separation of sister chromatids during meiosis II. (Ma et al., 2017)”

Regarding the use of the term “triplication”, we acknowledge the reviewer’s concern about potential ambiguity. To eliminate confusion, we now use the term “triplication” exclusively to describe the “duplication–inverted-triplication–duplication” structure (i.e., DUP-TRP/INV-DUP) in the revised manuscript.

R2.3 The distinct sex difference observed in the dnSV pattern is quite intriguing and adds an important dimension to the study. However, the derivation of the 10 kb size cutoff is not fully explained in the text (line 221-223). It would be helpful to provide more detail on how this cutoff was determined. Additionally, have you considered performing an enrichment test with different size cutoffs? Exploring this could yield further insights. Furthermore, the potential mechanisms behind these sex-specific patterns in dnSV generation are of great interest. If you have any hypotheses or proposed mechanisms, including them would greatly enrich the discussion. Finally, sharing the genomic positions of the dnSVs could be very beneficial for the scientific community, as it would facilitate independent validation in follow-up studies.

Thanks for making this relevant comment. In the revised manuscript, we compared paternal and maternal deletion size distribution rather than dichotomising size based on the arbitrary cutoff (i.e., 10 kb). In line with the original result, paternal deletions are enriched for smaller deletions whereas maternal deletions are enriched for larger deletions in both datasets (**Figure 4a**).

Figure 4a. | Comparison of the size of dnDELs according to parent of origin in GEL (left) and CEPH & SFARI cohort (right). The P values were calculated based on the two-sample Kolmogorov-Smirnov test.

We revised the sentence accordingly and also discussed the potential mechanisms behind these sex-specific patterns in dnSV generation as follows.

Line 241:

*“We observed a distinctive sex difference in patterns of dnSVs, specifically, maternal dnSVs were enriched for larger deletions, while paternal dnSVs were enriched for smaller deletions ($P = 4.99E-05$; **Figure 4A**). We further confirmed a similar enrichment pattern using an independent dataset (Belyeu et al., 2021) ($P = 1.63E-03$; **Figure 4A**). This gender-specific difference is potentially in line with a higher incidence of aneuploidy in oocytes than in sperm (Bell et al., 2020). The higher rate of aneuploidy is known to be associated with the distinct features of oocytes (Charalambous et al., 2023), such as the architecture of the meiotic spindle, the level of cortical tension at the oocyte surface, weaknesses in surveillance mechanisms that monitor chromosome segregation, and environmental factors.”*

Minor Revisions:

R2.5 A typo was noted in Figure 3d ("Inronic" should be "Intronic").

Thanks for spotting this. We corrected it.

R2.6 The significance marker in Figure S4b appears to be misaligned.

We corrected it.

R2.7 The term "read-based" in line 265 should likely be "read-depth."

Thanks for pointing this out. The reviewer is correct. Because we additionally called CNVs using a read-depth algorithm, we reorganised the paragraph.

Code review:

R2.8 Code block 1-510 is written in python syntax, 510-1224 is in R. It would be helpful to separate them into two files with proper file extension, e.g., .py and.

Thanks for this suggestion. Overall, we restructured the code repository and placed the scripts according to file extension and analysis type (https://github.com/hj6-sanger/GEL_SV/tree/main).

R2.9 suggests adding version for bedtools, samplot, and unfazed to help reproduce

We added the version for BEDTools (v2.31.0), Samplot (v1.3.0), and unfazed (v1.0.2)

.

R3.0 some internal scripts are not available for review:

Line 16, 'check.py' is not available

Line 36, 'bed.py' is not available

Line 54, 'stat.py' is not available

Thanks for spotting this. These scripts were used in filtering out SVs. We placed these scripts in https://github.com/hj6-sanger/GEL_SV/tree/main/Filtering and added more comments in these scripts.

R3.1 Description is missing for internal R functions, e.g., line 540, 578, 614, 676,766,798,848.

Apologies for the missing comments in the R scripts. We separated the R scripts and added a description and placed them in https://github.com/hj6-sanger/GEL_SV/tree/main/Analysis.

In summary, this manuscript presents a significant advancement in our understanding of dnSVs and their clinical implications. With minor revisions, I believe this study will make a valuable addition to the literature in genomics and rare disease research.

Reviewer #2 (Remarks on code availability):

- a. Code block 1-510 is written in python syntax, 510-1224 is in R. It would be helpful to separate them into two files with proper file extension, e.g., .py and .R
- b. suggests adding version for bedtools, samplot, and unfazed to help reproduce
- c. some internal scripts are not available for review:
Line 16, 'check.py' is not available
Line 36, 'bed.py' is not available
Line 54, 'stat.py' is not available
- d. Description is missing for internal R functions, e.g., line 540, 578, 614, 676,766,798,848

Reviewer #3 (Remarks to the Author):

Bibliography

- Bahrambeigi, V., Song, X., Sperle, K., Beck, C.R., Hijazi, H., Grochowski, C.M., Gu, S., Seeman, P., Woodward, K.J., Carvalho, C.M.B., Hobson, G.M., Lupski, J.R., 2019. Distinct patterns of complex rearrangements and a mutational signature of microhomeology are frequently observed in PLP1 copy number gain structural variants. *Genome Med.* 11, 80. <https://doi.org/10.1186/s13073-019-0676-0>.
- Beck, C.R., Carvalho, C.M.B., Akdemir, Z.C., Sedlazeck, F.J., Song, X., Meng, Q., Hu, J., Doddapaneni, H., Chong, Z., Chen, E.S., Thornton, P.C., Liu, P., Yuan, B., Withers, M., Jhangiani, S.N., Kalra, D., Walker, K., English, A.C., Han, Y., Chen, K., Lupski, J.R., 2019. Megabase length hypermutation accompanies human structural variation at 17p11.2. *Cell* 176, 1310-1324.e10. <https://doi.org/10.1016/j.cell.2019.01.045>.
- Bell, A.D., Mello, C.J., Nemes, J., Brumbaugh, S.A., Wysoker, A., McCarroll, S.A., 2020. Insights into variation in meiosis from 31,228 human sperm genomes. *Nature* 583, 259–264. <https://doi.org/10.1038/s41586-020-2347-0>.
- Belyeu, J.R., Brand, H., Wang, H., Zhao, X., Pedersen, B.S., Feusier, J., Gupta, M., Nicholas,

- T.J., Brown, J., Baird, L., Devlin, B., Sanders, S.J., Jorde, L.B., Talkowski, M.E., Quinlan, A.R., 2021. De novo structural mutation rates and gamete-of-origin biases revealed through genome sequencing of 2,396 families. *Am. J. Hum. Genet.* 108, 597–607. <https://doi.org/10.1016/j.ajhg.2021.02.012>.
- Brandler, W.M., Antaki, D., Gujral, M., Kleiber, M.L., Whitney, J., Maile, M.S., Hong, O., Chapman, T.R., Tan, S., Tandon, P., Pang, T., Tang, S.C., Vaux, K.K., Yang, Y., Harrington, E., Juul, S., Turner, D.J., Thiruvahindrapuram, B., Kaur, G., Wang, Z., Sebat, J., 2018. Paternally inherited cis-regulatory structural variants are associated with autism. *Science* 360, 327–331. <https://doi.org/10.1126/science.aan2261>.
- Carvalho, C.M.B., Coban-Akdemir, Z., Hijazi, H., Yuan, B., Pendleton, M., Harrington, E., Beaulaurier, J., Juul, S., Turner, D.J., Kanchi, R.S., Jhangiani, S.N., Muzny, D.M., Gibbs, R.A., Baylor-Hopkins Center for Mendelian Genomics, Stankiewicz, P., Belmont, J.W., Shaw, C.A., Cheung, S.W., Hanchard, N.A., Sutton, V.R., Lupski, J.R., 2019. Interchromosomal template-switching as a novel molecular mechanism for imprinting perturbations associated with Temple syndrome. *Genome Med.* 11, 25. <https://doi.org/10.1186/s13073-019-0633-y>.
- Carvalho, C.M.B., Lupski, J.R., 2016. Mechanisms underlying structural variant formation in genomic disorders. *Nat. Rev. Genet.* 17, 224–238. <https://doi.org/10.1038/nrg.2015.25>.
- Carvalho, C.M.B., Pehlivan, D., Ramocki, M.B., Fang, P., Alleva, B., Franco, L.M., Belmont, J.W., Hastings, P.J., Lupski, J.R., 2013. Replicative mechanisms for CNV formation are error prone. *Nat. Genet.* 45, 1319–1326. <https://doi.org/10.1038/ng.2768>.
- Carvalho, C.M.B., Pfundt, R., King, D.A., Lindsay, S.J., Zuccherato, L.W., Macville, M.V.E., Liu, P., Johnson, D., Stankiewicz, P., Brown, C.W., DDD Study, Shaw, C.A., Hurles, M.E., Ira, G., Hastings, P.J., Brunner, H.G., Lupski, J.R., 2015. Absence of heterozygosity due to template switching during replicative rearrangements. *Am. J. Hum. Genet.* 96, 555–564. <https://doi.org/10.1016/j.ajhg.2015.01.021>.
- Carvalho, C.M.B., Ramocki, M.B., Pehlivan, D., Franco, L.M., Gonzaga-Jauregui, C., Fang, P., McCall, A., Pivnick, E.K., Hines-Dowell, S., Seaver, L.H., Friebling, L., Lee, S., Smith, R., Del Gaudio, D., Withers, M., Liu, P., Cheung, S.W., Belmont, J.W., Zoghbi, H.Y., Hastings, P.J., Lupski, J.R., 2011. Inverted genomic segments and complex triplication rearrangements are mediated by inverted repeats in the human genome. *Nat. Genet.* 43, 1074–1081. <https://doi.org/10.1038/ng.944>.
- Charalambous, C., Webster, A., Schuh, M., 2023. Aneuploidy in mammalian oocytes and the impact of maternal ageing. *Nat. Rev. Mol. Cell Biol.* 24, 27–44. <https://doi.org/10.1038/s41580-022-00517-3>.
- Collins, R.L., Glessner, J.T., Porcu, E., Lepamets, M., Brandon, R., Lauricella, C., Han, L., Morley, T., Niestroj, L.-M., Ulirsch, J., Everett, S., Howrigan, D.P., Boone, P.M., Fu, J., Karczewski, K.J., Kellaris, G., Lowther, C., Lucente, D., Mohajeri, K., Nõukas, M., Talkowski, M.E., 2022. A cross-disorder dosage sensitivity map of the human genome. *Cell* 185, 3041–3055.e25. <https://doi.org/10.1016/j.cell.2022.06.036>.

- Du, H., Jolly, A., Grochowski, C.M., Yuan, B., Dawood, M., Jhangiani, S.N., Li, H., Muzny, D., Fatih, J.M., Coban-Akdemir, Z., Carlin, M.E., Scheuerle, A.E., Witzl, K., Posey, J.E., Pendleton, M., Harrington, E., Juul, S., Hastings, P.J., Bi, W., Gibbs, R.A., Liu, P., 2022. The multiple de novo copy number variant (MdnCNV) phenomenon presents with perizygotic DNA mutational signatures and multilocus pathogenic variation. *Genome Med.* 14, 122. <https://doi.org/10.1186/s13073-022-01123-w>.
- Grochowski, C.M., Bengtsson, J.D., Du, H., Gandhi, M., Lun, M.Y., Mehaffey, M.G., Park, K., Höps, W., Benito, E., Hasenfeld, P., Korbel, J.O., Mahmoud, M., Paulin, L.F., Jhangiani, S.N., Hwang, J.P., Bhamidipati, S.V., Muzny, D.M., Fatih, J.M., Gibbs, R.A., Pendleton, M., Carvalho, C.M.B., 2024. Inverted triplications formed by iterative template switches generate structural variant diversity at genomic disorder loci. *Cell Genomics* 4, 100590. <https://doi.org/10.1016/j.xgen.2024.100590>.
- Ishmukhametova, A., Chen, J.-M., Bernard, R., de Massy, B., Baudat, F., Boyer, A., Méchin, D., Thorel, D., Chabrol, B., Vincent, M.-C., Khau Van Kien, P., Claustres, M., Tuffery-Giraud, S., 2013. Dissecting the structure and mechanism of a complex duplication-triplication rearrangement in the DMD gene. *Hum. Mutat.* 34, 1080–1084. <https://doi.org/10.1002/humu.22353>.
- Kaplanis, J., Ide, B., Sanghvi, R., Neville, M., Danecek, P., Coorens, T., Prigmore, E., Short, P., Gallone, G., McRae, J., Genomics England Research Consortium, Carmichael, J., Barnicoat, A., Firth, H., O'Brien, P., Rahbari, R., Hurles, M., 2022. Genetic and chemotherapeutic influences on germline hypermutation. *Nature* 605, 503–508. <https://doi.org/10.1038/s41586-022-04712-2>.
- Kong, A., Frigge, M.L., Masson, G., Besenbacher, S., Sulem, P., Magnusson, G., Gudjonsson, S.A., Sigurdsson, A., Jonasdottir, Aslaug, Jonasdottir, Adalbjorg, Wong, W.S.W., Sigurdsson, G., Walters, G.B., Steinberg, S., Helgason, H., Thorleifsson, G., Gudbjartsson, D.F., Helgason, A., Magnusson, O.T., Thorsteinsdottir, U., Stefansson, K., 2012. Rate of de novo mutations and the importance of father's age to disease risk. *Nature* 488, 471–475. <https://doi.org/10.1038/nature11396>.
- Krumm, N., Sudmant, P.H., Ko, A., O'Roak, B.J., Malig, M., Coe, B.P., NHLBI Exome Sequencing Project, Quinlan, A.R., Nickerson, D.A., Eichler, E.E., 2012. Copy number variation detection and genotyping from exome sequence data. *Genome Res.* 22, 1525–1532. <https://doi.org/10.1101/gr.138115.112>.
- Liu, P., Yuan, B., Carvalho, C.M.B., Wuster, A., Walter, K., Zhang, L., Gambin, T., Chong, Z., Campbell, I.M., Coban Akdemir, Z., Gelowani, V., Writzl, K., Bacino, C.A., Lindsay, S.J., Withers, M., Gonzaga-Jauregui, C., Wiszniewska, J., Scull, J., Stankiewicz, P., Jhangiani, S.N., Lupski, J.R., 2017. An organismal CNV mutator phenotype restricted to early human development. *Cell* 168, 830-842.e7. <https://doi.org/10.1016/j.cell.2017.01.037>.
- Li, Y., Roberts, N.D., Wala, J.A., Shapira, O., Schumacher, S.E., Kumar, K., Khurana, E., Waszak, S., Korbel, J.O., Haber, J.E., Imielinski, M., Weischenfeldt, J., Beroukhi, R., Campbell, P.J., 2020. Patterns of somatic structural variation in human cancer genomes.

- Nature 578, 112–121. <https://doi.org/10.1038/s41586-019-1913-9>.
- Ma, R., Deng, L., Xia, Y., Wei, X., Cao, Y., Guo, R., Zhang, R., Guo, J., Liang, D., Wu, L., 2017. A clear bias in parental origin of de novo pathogenic CNVs related to intellectual disability, developmental delay and multiple congenital anomalies. *Sci. Rep.* 7, 44446. <https://doi.org/10.1038/srep44446>.
- Pfundt, R., Del Rosario, M., Vissers, L.E.L.M., Kwint, M.P., Janssen, I.M., de Leeuw, N., Yntema, H.G., Nelen, M.R., Lugtenberg, D., Kamsteeg, E.-J., Wieskamp, N., Stegmann, A.P.A., Stevens, S.J.C., Rodenburg, R.J.T., Simons, A., Mensenkamp, A.R., Rinne, T., Gilissen, C., Scheffer, H., Veltman, J.A., Hehir-Kwa, J.Y., 2017. Detection of clinically relevant copy-number variants by exome sequencing in a large cohort of genetic disorders. *Genet. Med.* 19, 667–675. <https://doi.org/10.1038/gim.2016.163>.
- Robinson, J.T., Thorvaldsdóttir, H., Winckler, W., Guttman, M., Lander, E.S., Getz, G., Mesirov, J.P., 2011. Integrative genomics viewer. *Nat. Biotechnol.* 29, 24–26. <https://doi.org/10.1038/nbt.1754>.
- Schuy, J., Grochowski, C.M., Carvalho, C.M.B., Lindstrand, A., 2022. Complex genomic rearrangements: an underestimated cause of rare diseases. *Trends Genet.* 38, 1134–1146. <https://doi.org/10.1016/j.tig.2022.06.003>.
- Turro, E., Astle, W.J., Megy, K., Gräf, S., Greene, D., Shamardina, O., Allen, H.L., Sanchis-Juan, A., Frontini, M., Thys, C., Stephens, J., Mapeta, R., Burren, O.S., Downes, K., Haimel, M., Tuna, S., Deevi, S.V.V., Aitman, T.J., Bennett, D.L., Calleja, P., Ouwehand, W.H., 2020. Whole-genome sequencing of patients with rare diseases in a national health system. *Nature* 583, 96–102. <https://doi.org/10.1038/s41586-020-2434-2>.
- Zuccherato, L.W., Alleva, B., Whitters, M.A., Carvalho, C.M.B., Lupski, J.R., 2016. Chimeric transcripts resulting from complex duplications in chromosome Xq28. *Hum. Genet.* 135, 253–256. <https://doi.org/10.1007/s00439-015-1614-x>.

RESPONSE TO REVIEWERS' COMMENTS

We appreciate again the time and thoughtful feedback provided by the reviewers on our manuscript, "Complex *de novo* structural variants are an underestimated cause of rare disorders".

REVIEWERS' COMMENTS

Reviewer #1 (Remarks to the Author):

These are excellent response letter and revised manuscript drafts. The authors addressed all of my previous concerns and clarified remaining questions. This is a beautiful and highly relevant manuscript for the rare disease community.

I have two additional minor comments:

1- There are several spelling errors in figures; please review them carefully. For example in Figure 1: reciprocal, Maternal, Translocation, etc. Also, there are instances where the word "triplication" is used but likely meant to be "duplication" (eg., statements in lines 139 and 399), please double-check.

Thanks for spotting our mistakes. We all revised them.

2-The authors should consider adding information about the number of the potential clinically relevant dnSVs in the abstract.

We added information regarding the number of the potentially clinically relevant dnSVs in the abstract as follows.

Abstract - Line:26

Among probands with dnSVs, 9% exhibit exon-disrupting pathogenic dnSVs associated with the probands' phenotype. Intriguingly, 12% of exon-disrupting pathogenic dnSVs and 22% of *de novo* deletions or duplications previously identified by array-based or exome-seq methods are found to be complex dnSVs.

Reviewer #2 (Remarks to the Author):

All my comments have been addressed.

Reviewer #3 (Remarks to the Author):
